

# Effect of sodium (*S*)-2-hydroxyglutarate in male, and succinic acid in female Wistar rats against renal ischemia-reperfusion injury, suggesting a role of the HIF-1 pathway

Eduardo Cienfuegos-Pecina[1], Tannya R. Ibarra-Rivera[2], Alma L. Saucedo[2], Luis A. Ramírez-Martínez[1], Deanna Esquivel-Figueroa[1], Ixel Domínguez-Vázquez[1], Karina J. Alcántara-Solano[1], Diana P. Moreno-Peña[1], Gabriela Alarcon-Galvan[3], Diana Raquel Rodríguez-Rodríguez[1], Liliana Torres-González[1], Linda E. Muñoz-Espinosa[1], Edelmiro Pérez-Rodríguez[4] and Paula Cordero-Pérez[1]

[1] Universidad Autonoma de Nuevo Leon, Liver Unit, Department of Internal Medicine, University Hospital "Dr. José E. González", Monterrey, Nuevo León, Mexico
[2] Universidad Autonoma de Nuevo Leon, Department of Analytical Chemistry, School of Medicine, Monterrey, Nuevo León, Mexico
[3] Universidad de Monterrey, Basic Science Department, School of Medicine, Monterrey, Nuevo León, Mexico
[4] Universidad Autonoma de Nuevo Leon, Transplant Service, University Hospital "Dr. José E. González", Monterrey, Nuevo León, Mexico

Corresponding author
Paula Cordero-Pérez,
paucordero@yahoo.com.mx

## ABSTRACT

**Background**. Ischemia–reperfusion (IR) injury is the main cause of delayed graft function in solid organ transplantation. Hypoxia-inducible factors (HIFs) control the expression of genes related to preconditioning against IR injury. During normoxia, HIF-$\alpha$ subunits are marked for degradation by the egg-laying defective nine homolog (EGLN) family of prolyl-4-hydroxylases. The inhibition of EGLN stabilizes HIFs and protects against IR injury. The aim of this study was to determine whether the EGLN inhibitors sodium (*S*)-2-hydroxyglutarate [(*S*)-2HG] and succinic acid (SA) have a nephroprotective effect against renal IR injury in Wistar rats.

**Methods**. (*S*)-2HG was synthesized in a 22.96% yield from commercially available L-glutamic acid in a two-step methodology (diazotization/alkaline hydrolysis), and its structure was confirmed by nuclear magnetic resonance and polarimetry. SA was acquired commercially. (*S*)-2HG and SA were independently evaluated in male and female Wistar rats respectively after renal IR injury. Rats were divided into the following groups: sham (SH), nontoxicity [(*S*)-2HG: 12.5 or 25 mg/kg; SA: 12.5, 25, or 50 mg/kg], IR, and compound+IR [(*S*)-2HG: 12.5 or 25 mg/kg; SA: 12.5, 25, or 50 mg/kg]; independent SH and IR groups were used for each assessed compound. Markers of kidney injury (BUN, creatinine, glucose, and uric acid) and liver function (ALT, AST, ALP, LDH, serum proteins, and albumin), proinflammatory cytokines (IL-1$\beta$, IL-6, and TNF-$\alpha$), oxidative stress biomarkers (malondialdehyde and superoxide dismutase), and histological parameters (tubular necrosis, acidophilic casts, and vascular congestion) were assessed. Tissue HIF-1$\alpha$ was measured by ELISA and Western blot, and the expression of Hmox1 was assessed by RT-qPCR.

**Results**. (S)-2HG had a dose-dependent nephroprotective effect, as evidenced by a significant reduction in the changes in the BUN, creatinine, ALP, AST, and LDH levels compared with the IR group. Tissue HIF-1α was only increased in the IR group compared to SH; however, (S)-2HG caused a significant increase in the expression of Hmox1, suggesting an early accumulation of HIF-1α in the (S)-2HG-treated groups. There were no significant effects on the other biomarkers. SA did not show a nephroprotective effect; the only changes were a decrease in creatinine level at 12.5 mg/kg and increased IR injury at 50 mg/kg. There were no effects on the other biochemical, proinflammatory, or oxidative stress biomarkers.

**Conclusion**. None of the compounds were hepatotoxic at the tested doses. (S)-2HG showed a dose-dependent nephroprotective effect at the evaluated doses, which involved an increase in the expression of Hmox1, suggesting stabilization of HIF-1α. SA did not show a nephroprotective effect but tended to increase IR injury when given at high doses.

## INTRODUCTION

Kidney transplantation is the definitive treatment for patients with end-stage chronic kidney disease. However, the shortage of available organs for transplantation is a formidable challenge. In Mexico, about five patients are on a waiting list for each available kidney (*CENATRA, 2018*). Despite the advances in knowledge about histocompatibility and immunosuppression, few steps have been taken to ameliorate the effects of ischemia–reperfusion (IR) injury during organ procurement. IR injury is one of the main causes of delayed graft function and is an important risk factor during surgical procedures such as thoracic, peripheral vascular, and major vascular surgery (*Chen & Date, 2015*; *Salvadori, Rosso & Bertoni, 2015*).

IR injury is a complex process that occurs when blood flow to an organ is impaired for a long period of time and then is suddenly restarted, perfusing the organ with oxygenated blood. This process is characterized by the release of reactive oxygen species (ROS), which involves several metabolic pathways such as the degradation of purine nucleotides (*Granger, 1988*; *McManaman & Bain, 2002*). During reperfusion, large amounts of oxygen enter the cellular microenvironment, where it leads to the production of superoxide radicals, HClO, and $H_2O_2$ (*Welbourn et al., 1991*). In addition, depletion of ATP during hypoxia causes alterations in mitochondrial function, inducing cytochrome c release, and apoptosis (*Dorweiler et al., 2007*).

Diverse strategies have been developed to prevent or limit IR injury, such as precondi-tioning drugs (hypoxia mimetics) and techniques such as ischemic preconditioning (IPC) (*Murry, Jennings & Reimer, 1986*) and remote ischemic preconditioning (RIPC) (*Przyklenk et al., 1993*). A key mechanism in IR injury prevention by IPC and RIPC appears to be the

upregulation of the family of hypoxia-inducible factors (HIFs) (*Randhawa, Bali & Jaggi, 2015*).

HIFs are a family of transcription factors that act as regulators of oxygen homeostasis in cells (*Semenza, 2014*). These transcription factors function as heterodimers whose structure includes an α-subunit, which is regulated by oxygen concentration, and a β-subunit, which is expressed constitutively and is not oxygen dependent (*Dengler, Galbraith & Espinosa, 2014*; *Kaelin & Ratcliffe, 2008*; *Wang & Semenza, 1995*). HIF family members stimulate the transcription of many genes, such as *HMOX1, EPO, VEGF, LDH*, and *PDK1* (*Semenza, 2014*). The most relevant of the HIF family is HIF-1α (*Wang & Semenza, 1995*). During normoxia, the α-subunit is hydroxylated at Pro-402 and Pro-564 residues by the egg-laying defective nine homolog (EGLN) family of prolyl-4-hydroxylases (*Ivan et al., 2001*; *Jaakkola et al., 2001*). EGLN-1 is the family member with the greatest affinity for HIF-1α (*Appelhoff et al., 2004*; *Karuppagounder & Ratan, 2012*). Hydroxylated α-subunits are ubiquitinated by the von Hippel–Lindau E3 ubiquitin ligase and degraded in the proteasome (*Ohh et al., 2000*). EGLN family members are α-ketoglutarate-dependent dioxygenases (*Bruick & McKnight, 2001*; *Jaakkola et al., 2001*), and inhibition of EGLN members stimulates HIF-1α accumulation (*Hill et al., 2008*). For that reason, these enzymes are considered to be a potential therapeutic target and have shown great promise in preclinical studies (*Bernhardt et al., 2009*; *Hill et al., 2008*).

A well-known EGLN inhibitor is 2-hydroxyglutarate (2HG), which was the first oncometabolite to be reported in the literature (*Dang et al., 2009*). Because of the structural analogy between α-ketoglutarate and 2HG, the latter possesses an inhibitory effect on several α-ketoglutarate-dependent dioxygenases, such as EGLN family members, increasing the intracellular concentration of HIF-1α (*Xu et al., 2011*). It has been shown that the (*S*)-2HG enantiomer has a stronger inhibitory activity than the (*R*)-2HG enantiomer, which can also act as a weak activator of EGLN-1 without promoting HIF-1α accumulation (*Koivunen et al., 2012*; *Xu et al., 2011*).

Several Krebs cycle metabolites also have EGLN-inhibitory activity. In vitro studies show that succinic acid (SA) inhibits EGLN family members because of its structural analogy with α-ketoglutarate and because of a possible product inhibition mechanism (*Koivunen et al., 2007*; *Myllyharju, 2009*), although additional data are needed to understand the role of SA in IR injury.

Given the reported activities and potential pharmacological properties of these compounds, the aim of this study was to determine whether the administration of EGLN inhibitors, (*S*)-2HG and SA, before induction of IR injury would have a nephroprotective effect in Wistar rats.

## MATERIALS & METHODS

### Sodium (*S*)-2HG synthesis

(*S*)-2HG was synthesized in two steps using a modification of previously reported methods (*Bauer & Gajewiak, 2004*; *Kolitz et al., 2009*). L-glutamic acid (Sigma-Aldrich, Saint Louis, MO, USA) was submitted to a diazotization reaction, which produced

**Figure 1** Sodium (*S*)-2-hydroxyglutarate synthesis.

(*S*)-5-oxotetrahydrofuran-2-carboxylic acid (MW: 130.10 g/mol), followed by alkaline hydrolysis, which produced sodium (*S*)-2HG (MW: 192.08 g/mol) (Fig. 1).

## (*S*)-5-oxotetrahydrofuran-2-carboxylic acid

A solution of 2.9426 g (20 mmol) of L-glutamic acid in 80 mL of 0.5 M $H_2SO_4$ was prepared and cooled to 0 °C in an ice bath. A solution of 8.2788 g (120 mmol) of $NaNO_2$ in 30 mL of double-distilled water was then added drop by drop. The reaction was performed at room temperature for 24 h with continuous stirring and then saturated with NaCl and extracted three times with 100 mL of ethyl acetate. The organic phase was recovered and dried with anhydrous $Na_2SO_4$. The solvent was low-pressure evaporated at 37 °C, and the synthesis product was purified in a silica column using an ethyl acetate mobile phase. Fractions of 5 mL were recovered and evaluated using thin layer chromatography with silica as the stationary phase and ethyl acetate as the mobile phase. The fractions containing the purified synthesis product were recovered, and the solvent was low-pressure evaporated at 37 °C.

## Sodium (*S*)-2HG

To synthesize (*S*)-2HG, 1.1329 g (8.7079 mmol) of (*S*)-5-oxotetrahydrofuran-2-carboxylic acid was dissolved in 100 mL of double-distilled water. A 10 M aqueous solution of NaOH was added drop by drop until a pH of 10 was reached, and the solution was subjected to continuous stirring for 2 h at room temperature. Most of the solvent was low-pressure coevaporated with ethanol at 37 °C. The reaction product was warmed in an oil bath, removed from the heat, and precipitated with anhydrous methanol. (*S*)-2HG crystals were recovered using a vacuum filtration system.

## Characterization by nuclear magnetic resonance

Nuclear magnetic resonance (NMR) data were acquired using a Bruker AVANCE III HD 400 MHz spectrometer (Bruker Corp., Billerica, MA, USA). A (S)-5-oxotetrahydrofuran-2-carboxylic acid sample was dissolved in $CDCl_3$ with 0.03% $v/v$ tetramethylsilane (Sigma-Aldrich) and transferred to 5 mm NMR standard tubes. (S)-2HG was dissolved in double-distilled water. The (S)-2HG aqueous solution was analyzed inside a 5 mm NMR tube using a coaxial system (Wilmad® coaxial insert, Wilmad-LabGlass, Vineland, NJ, USA), filled with $D_2O$ with 0.75% sodium 3-(trimethylsilyl)propionate-2,2,3,3-$d_4$ (TSP) (Sigma-Aldrich).

$^1H$ and $^{13}C$-NMR spectra of (S)-5-oxotetrahydrofuran-2-carboxylic acid were obtained using standard acquisition parameters. The (S)-2HG proton spectrum was acquired using the *noesypr1d* pulse sequence for water signal suppression. Compounds were identified by comparing the obtained data with previously reported spectra (*Bal & Gryff-Keller, 2002*). DEPT-135, HMBC, and HSQC spectra were also acquired. Relative (S)-2HG purity was determined by the integration of all the signals of the $^1H$-NMR spectrum, with exception of the $^{13}C$ couplings and the TSP signal using a method similar to published chromatographic procedures (*Pauli, Jaki & Lankin, 2007*). The obtained data were processed and analyzed using Bruker TopSpin 3.2 software (Bruker Corp.) and used to confirm (S)-2HG synthesis.

## Determination of Sodium (S)-2HG optical activity

The specific optical rotation ($[\alpha]_D^{20°C}$) of a 0.0944 g/mL aqueous solution of (S)-2HG was determined at 20 °C using the sodium D-line ($\lambda = 589$ nm) in a PerkinElmer 341 Polarimeter (PerkinElmer, Waltham, MA, USA) with a 50 mm cell.

## Animals

The animal procedures were performed according to the specifications of the Official Mexican Norm NOM-062-ZOO-1999. This project was approved by the Ethics and Research Committee of the School of Medicine, Universidad Autónoma de Nuevo León (Register number HI17-0002). Wistar rats, weighing 250–300 g, were used: 32 male rats to evaluate (S)-2HG activity and 42 females to evaluate SA activity. We randomly decided to use male or female rats for each assessed compound based on the availability of animals in our laboratory. Male and female rats exert a different behavior after induction of IR injury (*Robert et al., 2011*). Hence, we used independent control groups for each assessed compound. The rats were kept under standard conditions such as stable room temperature ($24 \pm 3$ °C) and 12 h light–dark cycle and had access to commercial rat pellets (Nutrimix de México, S.A. de C.V., Mexico City, Mexico) and water *ad libitum*.

## Experimental design
### Dose selection

To our knowledge, neither (S)-2HG nor SA has been tested in an in vivo model of renal IR injury. In vitro assays have demonstrated the accumulation of HIF-1α 6 h after the administration of (S)-2HG to cell cultures (*Xu et al., 2011*), and the administration of dimethyl fumarate (a structurally related EGLN-inhibitor) has demonstrated a hepatoprotective effect in Wistar at a dose of 25 mg/kg (*Takasu et al., 2017*). Hence, we decided to use the following experimental design:

*Evaluation of treatment with sodium (S)-2HG*

Rats were randomized and divided into the following groups.

1. Sham group (SH), $n = 4$: Rats were treated with double-distilled water administered *p.o.* twice per day for 2 days. They then underwent a laparotomy without induction of kidney IR injury and were allowed to recover for 15 h, after which they were sacrificed, and blood and kidney tissue samples were obtained.

2. Nontoxicity groups, $n = 4$ each: Rats were treated with (*S*)-2HG at a dose of 12.5 or 25 mg/kg (12.5Tox and 25Tox, respectively) in double-distilled water administered *p.o.* twice per day for 2 days. Eight hours after the final administration, the rats underwent the same procedure as the SH group.

3. IR group (IR), $n = 6$: Rats were treated with double-distilled water in the same way as for the SH group. They then underwent a laparotomy and induction of kidney IR injury comprising 45 min of ischemia and 15 h of reperfusion. After the reperfusion, the rats were sacrificed, and blood and kidney tissue samples were obtained.

4. IR+12.5 and IR+25 groups, $n = 6$ each: Rats were treated with (*S*)-2HG at a dose of 12.5 or 25 mg/kg in double-distilled water as for the nontoxicity groups. Eight hours after the final administration, rats underwent the same procedure as the IR group.

*Evaluation of treatment with SA*

Rats were randomized and divided into the following groups.

1. Sham group (SH), $n = 6$: Rats were treated with double-distilled water administered *p.o.* twice per day for 2 days. They then underwent a laparotomy without induction of kidney IR injury and 15 h of recovery. The animals were sacrificed, and blood and kidney tissue samples were obtained.

2. Nontoxicity groups, $n = 4$ each: Rats were treated with SA at a dose of 12.5, 25, or 50 mg/kg (12.5Tox, 25Tox, and 50Tox, respectively) in double-distilled water administered p.o. twice per day for 2 days. Eight hours after treatment, the rats received the same procedure as the SH group.

3. IR group (IR), $n = 6$: Rats were treated with double-distilled water as for the SH group. After treatment, they received a laparotomy with induction of kidney IR injury comprising 45 min of ischemia and 15 h of reperfusion. After reperfusion, the rats were sacrificed, and blood and kidney tissue samples were obtained.

4. IR+12.5, IR+25, and IR+50 groups, $n = 5$, 6, or 6 respectively: Rats with treated with SA at a dose of 12.5, 25, or 50 mg/kg in double-distilled water under the same conditions as for the nontoxicity groups. Eight hours after treatment, rats underwent the same procedure as the IR group.

## Induction of kidney IR injury

The procedure to induce IR injury was based on the protocol of *Torres-González et al. (2018)*. Rats were anesthetized by intraperitoneal injection with 100 mg/kg of ketamine (Anesket, PiSA Agropecuaria, S.A. de C.V. Reg. SAGARPA Q7833-028, Guadalajara, Jal., Mexico) and 10 mg/kg of xylazine (Sedaject, Vedilab S.A. de C.V. Reg. SAGARPA Q-0088-122, Querétaro, Qro., Mexico). After anesthesia, rats were shaved, and asepsis of the abdominal region was performed using Microdacyn antiseptic solution (Oculus

Technologies of Mexico, S.A. de C.V., Guadalajara, Jal., Mexico) followed by a 20% solution of chlorhexidine gluconate (Farmacéuticos Altamirano de México, S.A. de C.V., Mexico City, Mexico). A midline incision was then performed, both kidneys were exposed, the kidneys were dissected at both renal hila, and these structures were occluded using atraumatic vascular clamps for 45 min. After ischemia, the clamps were withdrawn, and the incision was sutured.

Rats were transferred to cages containing UV-sterilized sawdust, and tramadol dissolved in water was administered as an analgesic (50 mg/L, *ad libitum*) (Grünenthal GmbH, Stolberg, Germany). The reperfusion follow-up occurred over the next 15 h. The rats were then anesthetized by intraperitoneal injection of 50 mg/kg of ketamine and 5 mg/kg of xylazine. The incision was reopened, and 5–7 mL of blood was withdrawn by cava vein phlebotomy, which caused death by exsanguination. When the heart had stopped, both kidneys were removed. Half of each kidney was conserved in a phosphate-buffered 10% formalin solution (pH 7.4), and the other half was frozen at $-80\,°C$. Serum was separated from blood samples by centrifugation at 2,000 g for 12 min and stored at $-80\,°C$ until use.

## Analysis of biochemical markers, oxidative stress markers, and proinflammatory cytokines

To evaluate renal function, the serum concentrations of blood urea nitrogen (BUN) and creatinine were measured. To assess liver function, the serum activities of alanine aminotransferase (ALT), aspartate aminotransferase (AST), lactate dehydrogenase (LDH), and alkaline phosphatase (ALP) were measured. The serum concentrations of glucose (GLU), uric acid (UA), total proteins (TP), and albumin (ALB) were also measured. The biochemical analysis was performed using kinetic or end-point UV–visible spectrophotometric methods in an ILab Aries instrument (Instrumentation Laboratory SpA, Milan, Italy).

To assess the oxidative-stress-induced injury, the tissue concentration of malondialdehyde (MDA), and the tissue activity of total superoxide dismutase (SOD) was measured. Briefly, 200 mg of kidney tissue was homogenized, and the tissue homogenates were centrifuged three times at 10,000 g for 10 min at 4 °C. MDA and SOD quantification was performed using the supernatant.

MDA is one of the final products of lipid peroxidation, mainly arachidonic acid, and polyunsaturated fatty acids, and its activity was measured using the thiobarbituric acid colorimetric method with a thiobarbituric acid-reactive substances (TBARS) assay kit (Cayman Chemical Company, Ann Arbor, MI, USA). The product of this reaction was measured spectrophotometrically at 535 nm and normalized to the amount of homogenized tissue. SOD represents several metalloenzymes that form a crucial part of the cell enzymatic antioxidant defenses. SOD catalyzes the dismutation of the superoxide anion to $O_2$ and $H_2O_2$. Total SOD activity was measured using a method of inhibition of the reduction of Dojindo's water-soluble tetrazolium salt [WST-1; 2-(4-iodophenyl)-3-(4-nitrophenyl)-5-(2,4-disulfophenyl)-2H tetrazolium, monosodium salt] in the presence of xanthine oxidase, xanthine, and oxygen (Sigma-Aldrich). SOD activity ameliorates the reduction in tetrazolium and was measured spectrophotometrically at 450 nm.

The serum concentrations of the proinflammatory cytokines interleukin 1 beta (IL-1β), interleukin 6 (IL-6), and tumor necrosis factor alpha (TNF-α) were measured using a commercial sandwich enzyme-linked immunosorbent assay (ELISA) (Pepro-Tech, Mexico City, Mexico). Briefly, 96-well microplates were covered with rabbit antibodies specific to the measured cytokine. Plates were washed with phosphate-buffered saline containing 0.05% Tween 20 (pH 7.2) and then blocked with 1% serum bovine albumin. Samples and standards were added and detected using specific biotinylated detection antibodies and avidin-conjugated horseradish peroxidase (HRP). 2,2-Azino-bis(3-ethylbenzothiazoline-6-sulfonic) acid was used as the substrate for HRP; the reaction produced a green chromogen whose concentration was proportional to the concentration of the evaluated cytokine. The concentration of the end-product was measured spectrophotometrically at 405 nm, with an additional 650 nm wavelength measurement for correction.

## Renal histopathology evaluation

Fixed kidney tissue was paraffin embedded and processed using standard histological techniques. Paraffin blocks were cut using a microtome at a thickness of 4 μm. Sections were deparaffinized, hydrated, stained with hematoxylin–eosin, and evaluated microscopically with the assessor blinded to the identity of the groups.

The presence of tissue damage indicators, such as tubular necrosis, acidophilic casts, and vascular congestion, was assessed. The damage was reported according to the semiquantitative scale published by *Kobuchi et al. (2009)* as follows: 0 = no damage; 1 = mild damage (unicellular patchy isolated damage); 2 = moderate damage (<25%); 3 = severe damage (25–50%) and 4 = very severe damage (>50%).

## Tissue HIF-1α concentration measurement
### ELISA

HIF-1α concentration was assessed in tissue homogenates as described for oxidative stress biomarkers. The analysis was performed using a sandwich ELISA method (Sigma-Aldrich) with specific capture and detection antibodies, the latter conjugated to HRP. 3,3′,5,5′-Tetramethylbenzidine was used as the substrate, and the reaction was stopped by the addition of a 0.2 M solution of $H_2SO_4$. Absorbance was measured at 450 nm. HIF-1α concentration is reported relative to the amount of homogenate tissue.

### Western blot

HIF-1α was measured in kidney tissue by a Western blot methodology. Briefly, pools of 100 mg of kidney tissue were mechanically homogenized in 500 μL of 1X lysis buffer containing 10 mM Tris-HCl (pH 7.5); 50 mM KCl, 2 mM $MgCl_2$, 1% Triton X-100, 1 mM dithiothreitol, 1 mM phenylmethylsulfonyl fluoride and cOmplete™ Protease Inhibitor Cocktail according to manufacturer conditions (Roche, Basilea, Switzerland) at 4 °C and centrifuged at 13,000 g for 5 min. Proteins were quantified in the supernatant by the Bradford method (Bio-Rad, Hercules, CA, USA) and 150 μg of protein was separated on 12% polyacrylamide gels. Proteins were transferred onto PVDF membranes and monoclonal mouse antibodies were used for the detection of HIF-1α (sc-13515; 1:300, 6.7 μg/mL, Santa Cruz Biotechnology, Dallas, TX, USA) and GAPDH (MAB5718; 1:10,000;

0.05 μg/mL; R&D Systems, Minneapolis, MN, USA). A secondary HRP-conjugated anti-mouse antibody was used (W4021; 1:10,000; 0.1 μg/mL; Promega, Madison, WI, USA), and the signal was measured by chemiluminescence using a Clarity[TM] Western ECL Substrate (Bio-Rad).

## Quantitative RT-PCR

Total RNA was extracted from 100 mg tissue (pool of both kidneys) using TRIzol reagent (Invitrogen, Thermo Fisher Scientific, Carlsbad, CA, USA) according to the manufacturer's specifications. RNA samples were precipitated with 100% isopropanol, washed with 75% ethyl-alcohol, resuspended in RNase-free water, quantified using a Microdrop Multiskan GO (Thermo Fisher Scientific, Carlsbad, CA, USA) and stored at −80 °C. RT-qPCR was performed with GoTaq 1-Step (Promega) using as template 100 ng RNA to quantify gene expression levels of Hmox1 and β-actin using the following oligonucleotides: Hmox1 forward 5′-GCCTGCTAGCCTGGTTCAAGA-3′, Hmox1 reverse 5′-GAGTGTGAGGACCCATCGCA-3′, β-actin forward 5′-CCCTGGCTCCTAGCACCAT-3′ and β-actin reverse 5′-GATAGAGCCACCAATCCACACA-3′. Primers were used at a concentration of 100 nM. For each PCR reaction, 10 μL of GoTaq qPCR Master Mix 2X, 0.4 μL Go Script RT Mix 50X, and 0.33 μL CXR Reference Dye were used, to complete a total volume of 20 μL. Thermal cycling conditions were as follows: reverse transcription one cycle to 37 °C for 15 min, reverse transcription inactivation and Go Taq DNA Polymerase activation one cycle to 95 °C for 10 min, followed by 40 cycles of 95 °C for 10 s and 60 °C for 30 s. β-actin was used as the housekeeping gene and the fold changes of gene expression were calculated by the $2^{-\Delta\Delta Ct}$ method.

## Statistical analysis

Data were analyzed using a one-way analysis of variance followed by the Tukey post hoc test or by the Kruskal–Wallis nonparametric test followed by the Dunn post hoc test, depending on the data distribution. Data expressed as fold changes were transformed logarithmically before the statistical analysis. The analysis was performed using GraphPad Prism software (v. 7.0; GraphPad, San Diego, CA, USA). The results are expressed as mean ±standard deviation (SD) or median (interquartile range), depending on the data distribution. Differences between means were considered to be significant at $p < 0.05$.

# RESULTS

## Sodium (*S*)-2-HG synthesis and chemical characterization
### (S)-5-Oxotetrahydrofuran-2-carboxylic acid

After chromatographic purification, 1.9012 g (14.6134 mmol) of (*S*)-5-oxotetrahydrofuran-2-carboxylic acid was obtained at a 73.07% yield, as a yellow oil, which spontaneously crystallized and produced beige crystals, which were poorly soluble in water but soluble in ethyl acetate, methanol, and ethyl ether.

[1]H-NMR (MeOD): 4.994 ppm (2.8 H, m); 2.574 ppm (3H, m); and 2.296 ppm (1H, m). The signal at 4.994 ppm overlapped partially with the signal of the residual water absorbed by the solvent. [13]C-NMR (MeOD): 179.20, 173.58, 77.52, 27.94, and 27.08 ppm.

Both $^{1}$H and $^{13}$C-NMR spectra are consistent with the values reported previously (*Bal & Gryff-Keller, 2002*).

### Sodium (S)-2HG

After precipitation, 882.2 mg (4.5929 mmol) of (*S*)-2HG was obtained at a 31.43% yield as a beige crystalline solid, which was highly hygroscopic and was soluble in water but insoluble in methanol, ethanol, acetone, ethyl acetate, ethyl ether, or hexane. The global yield of the synthesis of (*S*)-2HG from L-glutamic acid was 22.96%.

In $^{1}$H-NMR, the following four signals were observed (Fig. S1): 4.004 ppm (1H, dd; $J'$ = 7.6 Hz, $J''$ = 4 Hz); 2.223 ppm (2H, m); 1.960 ppm (1H, m); and 1.827 ppm (1H, m). In $^{13}$C-NMR, five signals were observed (Fig. S2): 185.71, 184.10, 75.01, 36.38, and 33.92 ppm. The DEPT-135 experiment showed only three signals: 75.01, 36.38, and 33.92 ppm (Fig. S3). The HSQC (Fig. S4) and HMBC (Fig. S5) experiments allowed the unequivocal signal assignation of the (*S*)-2HG molecule, which coincided with data reported in the literature (*Bal & Gryff-Keller, 2002*). (*S*)-2HG was obtained with a 98.6% relative NMR purity.

A value of $[\alpha]_{D}^{20°C} = -8.40°$ cm$^{3}$ g$^{-1}$ dm$^{-1}$ was observed, which coincides with data previously reported (*Ritthausen, 1872*; *Sigma-Aldrich, 2018*).

## Evaluation of the effects of sodium (S)-2HG treatment
### Kidney and liver injury biomarkers

To examine whether (*S*)-2HG had any nephroprotective effects, we compared the SH, 12.5+IR, 25+IR, and IR groups. Treatment with (*S*)-2HG produced a dose-dependent decrease in the serum concentrations of BUN and creatinine and increased the TP concentration compared to the IR group. Similarly, it decreased the serum activities of AST, ALP, and LDH compared to the IR group. There were no significant differences between groups in serum concentrations of ALB, GLU, and UA, or the serum activity of ALT (Fig. 2, Table 1, Data S1).

Because one of our aims was to identify any evidence of hepato- or nephrotoxic effects caused by the administration of (*S*)-2HG, we compared the nontoxicity and SH groups. There were no significant differences between the 12.5Tox, 25Tox, and SH groups in the serum concentration of BUN or creatinine. The serum concentrations of ALB, GLU, TP, and UA also did not differ between groups. There were no significant differences between the serum activities of ALT, AST, ALP or LDH (Fig. 2, Table 1, Data S1).

### Biomarkers of Oxidative Stress

The total % inhibition rate of WST-1 reduction caused by SOD activity and the normalized tissue MDA concentration did not differ significantly between groups (Table 1).

### Proinflammatory Cytokines

The serum concentrations of IL-1β, IL-6, and TNF-α did not differ significantly between groups (Table 1).

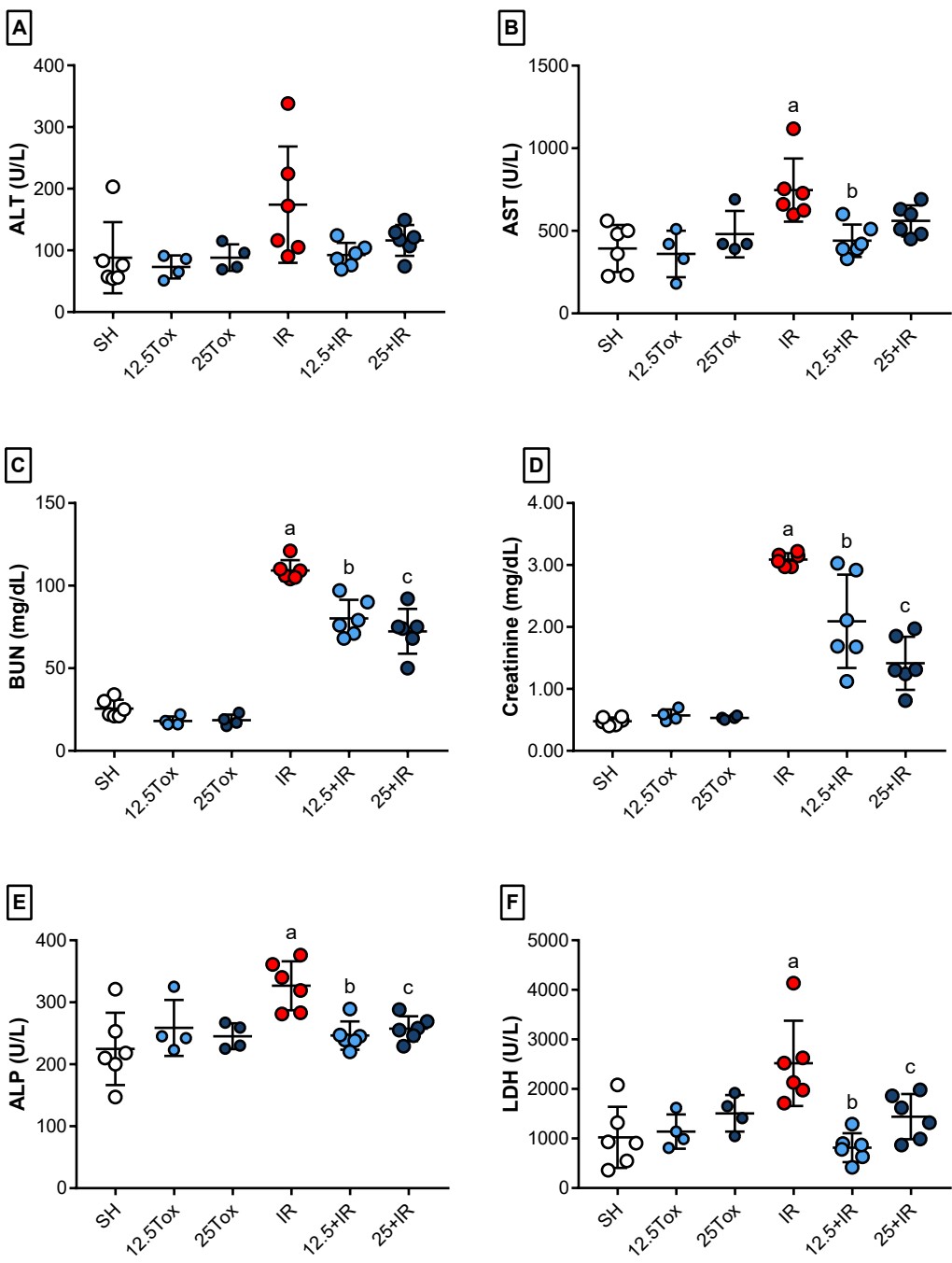

**Figure 2 Biomarkers of kidney and liver injury after administration of sodium (S)-2-hydroxyglutarate.** Values are expressed as mean ± SD. The comparisons were between the 12.5Tox, 25Tox, and SH groups, and between the SH, 12.5+IR, 25+IR, and IR groups. One-way ANOVA test, Tukey post hoc test. (A) Effect of (S)-2HG treatment on the serum activity of ALT. (B) Effect of (S)-2HG treatment on the serum activity of AST; a) $p = 0.0018$ vs. SH group; b) $p = 0.0081$ vs. IR group. (C) Effect of (S)-2HG treatment on the serum concentration of (continued on next page...)

**Figure 2 (...continued)**
BUN; a) $p < 0.0001$ vs. SH group; b) $p < 0.0001$ vs. IR group; c) $p < 0.0001$ vs. IR group. (D) Effect of ($S$)-2HG treatment on the serum concentration of creatinine; a) $p < 0.0001$ vs. SH group; b) $p = 0.0017$ vs. IR group; c) $p < 0.0001$ vs. IR group. (E) Effect of ($S$)-2HG treatment on the serum activity of ALP; a) $p = 0.0010$ vs. SH group; b) $p = 0.0119$ vs. IR group; c) $p$ 0.0394 vs. IR group. (F) Effect of ($S$)-2HG treatment on the serum activity of LDH; a) $p = 0.0009$ vs. SH group; b) $p = 0.0002$ vs. IR group; c) $p = 0.0240$ vs. IR group. ($S$)-2HG = sodium ($S$)-2-hydroxyglutarate; ALT = alanine aminotransferase; AST = aspartate aminotransferase; BUN = blood urea nitrogen; ALP = alkaline phosphatase; LDH = lactate dehydrogenase.

**Table 1   Biomarkers of kidney and liver injury, oxidative stress biomarkers, and proinflammatory cytokines after administration of ($S$)-2-hydroxyglutarate.**

| Biomarker | SH | 12.5Tox | 25Tox | IR | 12.5+IR | 25+IR |
|---|---|---|---|---|---|---|
| ALB (g/dL) | $3.0 \pm 0.1$ | $3.1 \pm 0.1$ | $3.1 \pm 0.2$ | $3.0 \pm 0.2$ | $3.0 \pm 0.1$ | $3.1 \pm 0.2$ |
| GLU (mg/dL) | $164 \pm 29$ | $193 \pm 31$ | $181 \pm 33$ | $134 \pm 37$ | $144 \pm 25$ | $134 \pm 24$ |
| TP (g/dL) | $5.3 \pm 0.2$ | $5.4 \pm 0.3$ | $5.8 \pm 0.3$ | $5.1 \pm 0.3$ | $5.2 \pm 0.2$ | $5.6 \pm 0.2$ * |
| UA (mg/dL) | $0.6 \pm 0.1$ | $0.5 \pm 0.1$ | $0.6 \pm 0.2$ | $0.4 \pm 0.3$ | $0.4 \pm 0.1$ | $0.6 \pm 0.1$ |
| SOD (%) | $93 \pm 2$ | $91 \pm 1$ | $93 \pm 7$ | $95 \pm 3$ | $97 \pm 5$ | $95 \pm 6$ |
| MDA (μmol/g) | $75.48 \pm 25.75$ | $80.00 \pm 32.04$ | $70.63 \pm 24.12$ | $87.36 \pm 33.91$ | $86.13 \pm 25.48$ | $73.57 \pm 13.66$ |
| IL-1β (ng/mL) | $1.50 \pm 0.67$ | $1.50 \pm 0.41$ | $1.34 \pm 0.94$ | $2.19 \pm 1.57$ | $1.70 \pm 1.21$ | $1.51 \pm 0.73$ |
| IL-6 (ng/mL) | $3.44 \pm 1.26$ | $4.00 \pm 0.65$ | $2.91 \pm 1.33$ | $3.05 \pm 2.49$ | $2.49 \pm 0.45$ | $2.42 \pm 0.73$ |
| TNF-α (ng/mL) | $0.78 \pm 0.35$ | $0.85 \pm 0.34$ | $1.02 \pm 0.92$ | $1.06 \pm 1.12$ | $0.61 \pm 0.55$ | $0.71 \pm 0.41$ |

Notes.
One-way ANOVA test, Tukey post hoc test. The comparisons were between the 12.5Tox, 25Tox, and SH groups, and between the SH, 12.5+IR, 25+IR, and IR groups.
ALB, albumin; ALP, alkaline phosphatase; GLU, glucose; IL-1β, interleukin 1β; IL-6, interleukin 6; LDH, lactate dehydrogenase; MDA, malondialdehyde; SOD, superoxide dismutase; TNF-α, tumor necrosis factor α; TP, total proteins; UA, uric acid.
*$p = 0.0358$ vs. IR group.

### Evaluation of renal histopathology

Renal histopathology parameters did not differ significantly between the SH, 12.5Tox, and 25Tox groups. The severity of tubular necrosis and the presence of acidophilic casts differed significantly between the IR and SH groups (Table 2, representative microphotographs are shown in Fig. 3).

### Evaluation of the effect of sodium (S)-2HG treatment on the tissue HIF-1α concentration

To assess whether the nephroprotective effect of ($S$)-2HG was caused by a promotion of the accumulation of HIF-1α, this protein was measured in kidney tissue homogenates. ELISA assessment showed that the normalized HIF-1α tissue concentration was significantly higher in the 12.5Tox and IR groups compared to the SH group. The normalized HIF-1α tissue concentration was significantly lower in the 25+IR group compared with the IR group. The normalized HIF-1α tissue concentration did not differ significantly between the 25Tox and 12.5+IR groups compared with the SH and IR groups, respectively (Fig. 4).

Western blot analysis showed no significative differences in the HIF-1α concentration in the groups Tox12.5 and Tox25 compared to the SH group. However, a 3.434-fold increase in the concentration of HIF-1α in the IR group compared to the SH group was
**Table 2  Evaluation of renal histopathology after administration of (*S*)-2-hydroxyglutarate.**

| Parameter | SH | 12.5Tox | 25Tox | IR | 12.5+IR | 25+IR |
|---|---|---|---|---|---|---|
| Tubular Necrosis | 0.00 (0.00–0.00) | 0.00 (0.00–0.75) | 0.00 (0.00–0.00) | 4.00 (4.00–4.00)[†] | 2.50 (1.75–3.00) | 2.50 (1.75–3.00) |
| Acidophilic Casts | 0.00 (0.00–0.50) | 0.50 (0.00–1.75) | 0.00 (0.00–0.00) | 4.00 (3.00–4.00)[‡] | 3.00 (1.75–3.25) | 2.00 (2.00–2.25) |
| Vascular Congestion | 1.00 (0.00–1.25) | 0.50 (0.00–1.75) | 0.50 (0.00–1.00) | 2.00 (1.00–4.00) | 2.00 (1.75–2.00) | 1.00 (1.00–2.00) |

**Notes.**

Kruskal–Wallis test, Dunn post hoc test. Values expressed as median (interquartile range). The comparisons were between the 12.5Tox, 25Tox, and SH groups, and between the SH, 12.5+IR, 25+IR, and IR groups. Significant differences when comparing IR vs. SH group.

[†] $p = 0.0002$.

[‡] $p = 0.0016$.

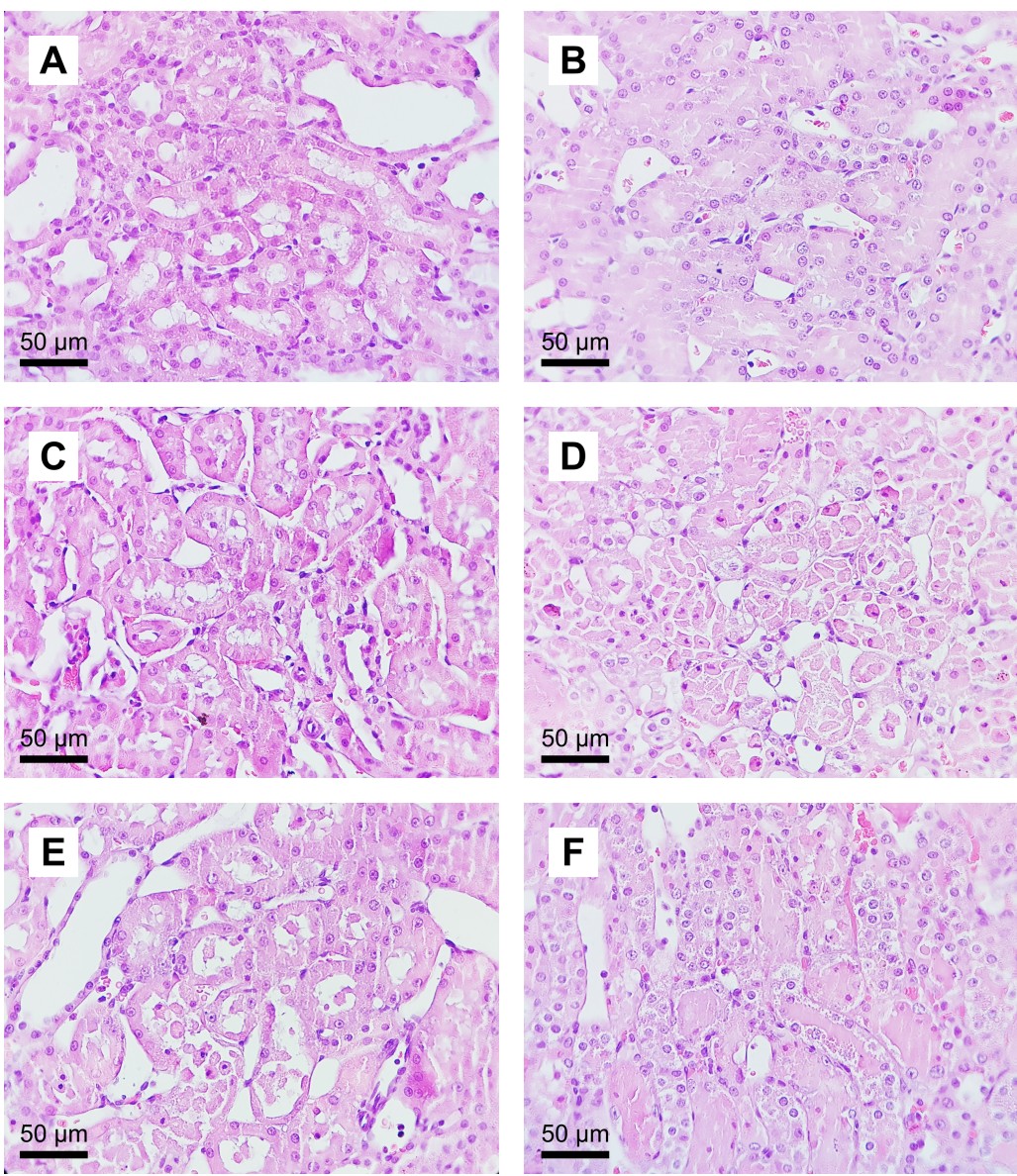

**Figure 3  Microphotographs of renal histology in the sodium ($S$)-2-hydroxyglutarate-treated groups.** Hematoxylin and eosin staining (original magnification × 400). Conserved tissue architecture was observed in (A) the SH group; (B) the 12.5Tox group; and (C) the 25Tox group. Diffuse and severe tubular necrosis was observed in (D) the IR group, and focal tubular necrosis in (E) the 12.5+IR group and (F) the 25+IR group.

observed. On the other hand, the groups 12.5+IR and 25+IR showed a significantly lower concentration of HIF-1α compared to the IR group (Fig. 5).

### Evaluation of the effect of sodium (S)-2HG on the expression of Hmox1 in kidney tissue

The expression of Hmox1 was significantly increased after the treatment with ($S$)-2HG at a dose of 25 mg/kg (14.15-fold from the SH group). Treatment with a dose of 12.5 mg/kg

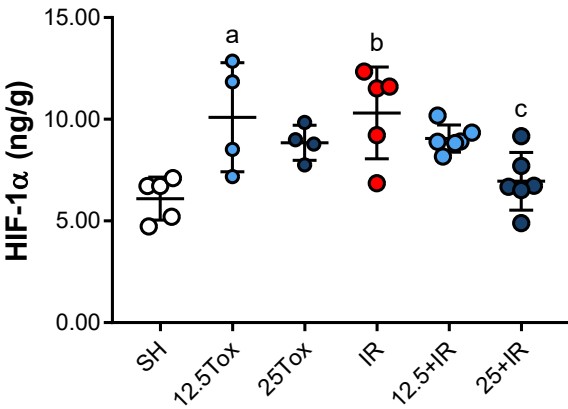

**Figure 4** **Tissue HIF-1α concentrations measured by ELISA after administration of sodium (*S*)-2-hydroxyglutarate.** Values are expressed as mean ± SD. The comparisons were between the 12.5Tox, 25Tox, and SH groups, and between the SH, 12.5+IR, 25+IR, and IR groups. One-way ANOVA test, Tukey post hoc test. a) $p = 0.0114$ vs. SH group; b) $p = 0.0040$ vs. SH group; c) $p = 0.0211$ vs. IR group. HIF-1α = hypoxia-inducible factor 1 alpha.

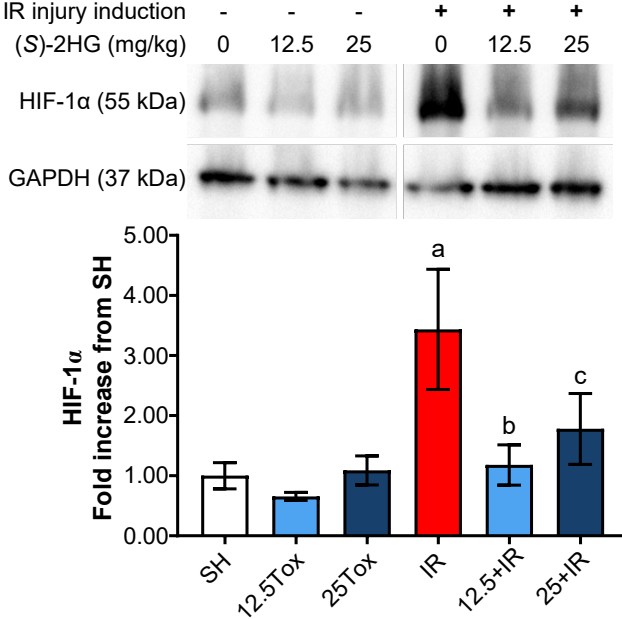

**Figure 5** **Tissue HIF-1α expression assessed by Western blot after administration of sodium (*S*)-2-hydroxyglutarate.** Values are expressed as mean ± SD. The comparisons were between the 12.5Tox, 25Tox, and SH groups, and between the SH, 12.5+IR, 25+IR, and IR groups. One-way ANOVA test, Tukey post hoc test (data were logarithmically transformed prior to the statistical analysis). a) $p < 0.0001$ vs. SH group; b) $p = 0.0001$ vs. IR group; c) $p = 0.0151$ vs. IR group. HIF-1α = hypoxia-inducible factor 1 alpha.

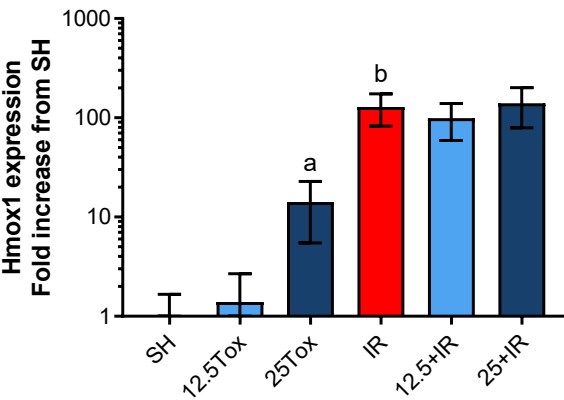

**Figure 6** **Tissue expression of Hmox1 assessed by RT-qPCR after administration of (*S*)-2-hydroxyglutarate.** Values are expressed as mean ± SD. The comparisons were between the 12.5Tox, 25Tox, and SH groups, and between the SH, 12.5+IR, 25+IR, and IR groups. One-way ANOVA test, Tukey post hoc test (data were logarithmically transformed prior to the statistical analysis). a) $p = 0.0011$ vs. SH group; b) $p < 0.0001$ vs. SH group. Hmox1 = heme oxygenase 1.

did not have a significant effect on its expression (Fig. 6). The induction of IR injury in the IR, 12.5+IR, and 25+IR groups drastically increased the expression of Hmox1 compared to the SH group (128.42-fold, 98.94-fold, and 140.01-fold, respectively; Fig. 6).

## Evaluation of the effects of SA treatment
### *Kidney and liver injury biomarkers*
To identify any nephroprotective effects caused by the administration of SA, we compared the SH, 12.5+IR, 25+IR, and 50+IR groups with the IR group. The administration of SA did not decrease BUN compared to the IR group but slightly decreased creatinine concentration in the 12.5+IR group. There were no significative differences in the creatinine concentrations between the 25+IR and 50+IR compared with the IR group. There was no significant difference between groups in the serum concentrations of ALB, TP, or UA, or in the serum activities of ALT, AST, ALP, or LDH (Fig. 7, Table 3, Data S2).

The 12.5Tox, 25Tox, and 50Tox groups were compared with the SH group to identify any hepato- or nephrotoxic effects caused by the administration of SA. There were no significant differences between groups in the serum concentrations of BUN, creatinine, ALB, GLU, TP, or UA. There were also no significant differences between groups in the serum activities of ALT, AST, ALP, or LDH (Fig. 7, Table 3, Data S2).

### *Biomarkers of oxidative stress*
The total % inhibition rate of WST-1 reduction caused by SOD activity and the normalized tissue MDA concentration did not differ significantly between groups (Table 3).

### *Proinflammatory cytokines*
The serum concentrations of IL-1β, IL-6, and TNF-α did not differ significantly between groups (Table 3).

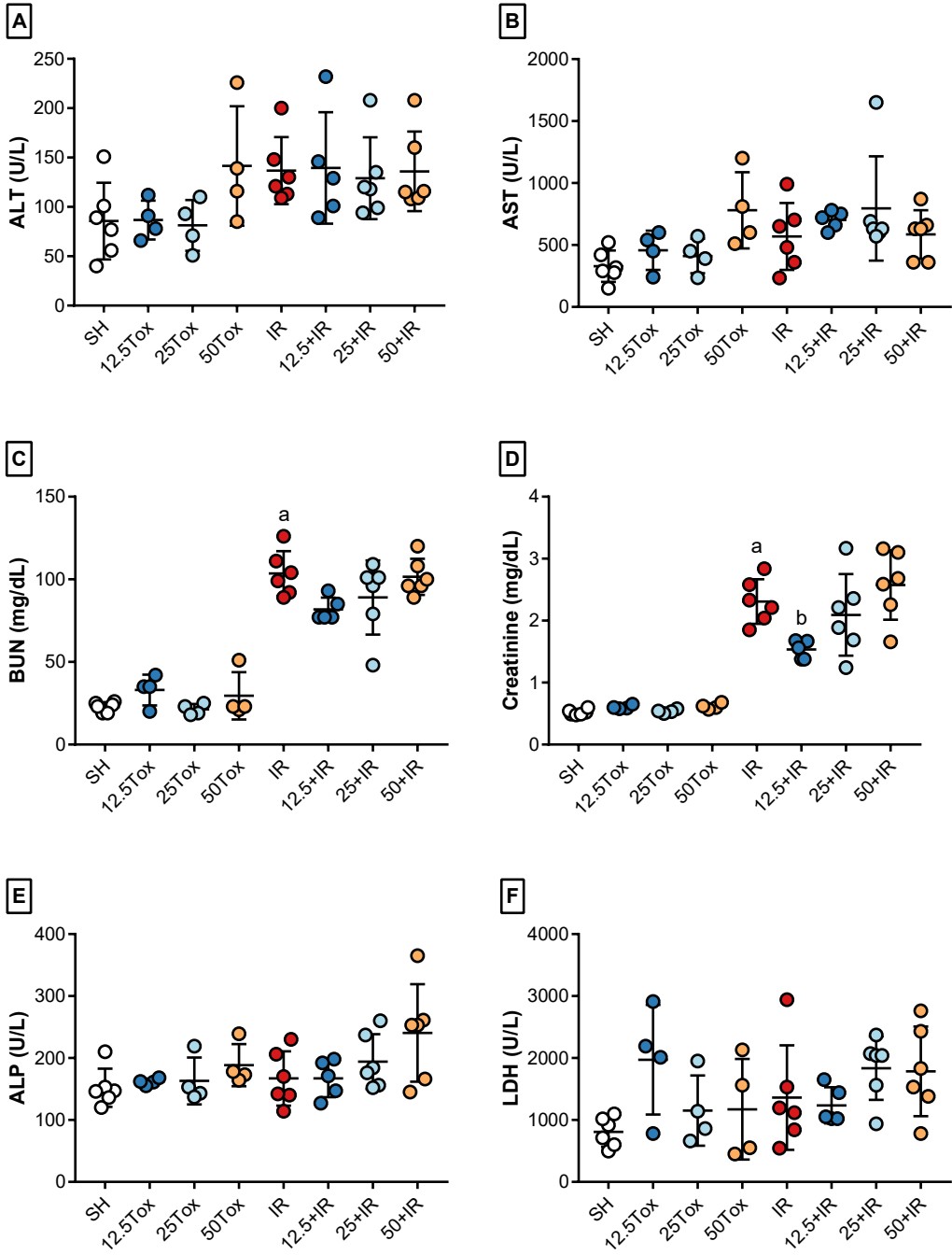

**Figure 7 Biomarkers of kidney and liver injury after administration of succinic acid.** Values are expressed as mean ± SD. The comparisons were between the 12.5Tox, 25Tox, 50Tox, and SH groups, and between the SH, 12.5+IR, 25+IR, 50+IR, and IR groups. One-way ANOVA test, Tukey post hoc test. (A) Effect of SA treatment on the serum activity of ALT. (B) Effect of SA treatment on the serum activity of AST. (C) Effect of SA treatment on the serum concentration of BUN; a) $p < 0.0001$ vs. SH group. (D) Effect of SA treatment on the serum concentration of creatinine; a) $p < 0.0001$ vs. SH group; (b) $p = 0.0288$ vs. IR group. (E) Effect of SA treatment on the serum activity of ALP. (F) Effect of SA treatment on the serum activity of LDH. SA = succinic acid; ALT = alanine aminotransferase; AST = aspartate aminotransferase; BUN = blood urea nitrogen; ALP = alkaline phosphatase; LDH = lactate dehydrogenase.

### *Evaluation of renal histopathology*

Tubular necrosis was significantly increased in the IR group compared with the SH group. None of the other parameters evaluated differed significantly between groups (Table 4, representative microphotographs are shown in Fig. 8).

## DISCUSSION

Despite improvements in the prevention of immunological graft rejection, few steps have been taken to decrease the magnitude of IR injury. For that reason, strategies are needed to ameliorate the deleterious effects of IR injury. Several mechanisms are involved in the processes causing IR injury, such as ROS generation, redox imbalance, necrosis–apoptosis, and the inflammatory response (*Dorweiler et al., 2007*). To ameliorate the effects of injury, various strategies have been tried including the use of antioxidants (*Li et al., 2014*; *Torres-González et al., 2018*; *Yildiz et al., 2015*), anti-inflammatory compounds (*Zhang et al., 2018*; *Zhu et al., 2014*), and surgical strategies, such as IPC (*Cai et al., 2013*; *Cai et al., 2008*; *Shen et al., 2018*) and RIPC (*Albrecht et al., 2013*; *Hausenloy & Yellon, 2007*; *Kalakech et al., 2013*).

In this study, we evaluated the ability of pharmacological preconditioning to prevent or reduce the effects of IR injury. Previous studies have demonstrated significant amelioration of IR injury using inhibitors of the EGLN family (*Hill et al., 2008*; *Milkiewicz, Pugh & Egginton, 2004*). Therefore, we evaluated the effects of two structural analogs of α-ketoglutarate. (*S*)-2HG was synthesized by a diazotization reaction, followed by alkaline hydrolysis. The diazotization of α-amino acids is considered to be a classic reaction for synthesizing α-hydroxy acids through the enantioselective route (*Brewster et al., 1950*; *Neuberger, 1948*; *Zollinger, 1995*) involving a double-$S_N2$ mechanism on the α carbon (*Markgraf & Davis, 1990*; *Pleissner, Wimmer & Eriksen, 2011*; *Van Draanen & Hengst, 2010*; *Zollinger, 1995*). In this work, we obtained (*S*)-2HG from L-glutamic acid in a two-step procedure. After the diazotization reaction, the (*S*)-2-hydroxyglutaric lactone was produced; in a second step, the alkaline hydrolysis furnished the desired (*S*)-2HG (*Williams et al., 2006*). The chemical identity of (*S*)-2HG was established by comparing the experimental NMR spectra and optical activity with those reported in the literature (*Bal & Gryff-Keller, 2002*). The optical activity of (*S*)-2HG was measured and determined as $[\alpha]_D^{20°C} = -8.40°$ $cm^3 g^{-1} dm^{-1}$, which is similar to the reported value of $-8.5 \pm 1.5°$ $cm^3 g^{-1} dm^{-1}$ (*Ritthausen, 1872*; *Sigma-Aldrich, 2018*). These data confirm that the synthetic product was (*S*)-2HG.

Several animal models have been used to study renal acute kidney injury caused by IR. Initial studies were performed in animals of relatively large size, such as dogs, pigs, and rabbits, and since the 1960s rat models have been one of the most reported in the literature (*Wei & Dong, 2012*). However, in recent years, mouse models have become popular mainly due to the availability of standardized genetically engineered strains. Rat models of IR injury have been well characterized (*Heyman et al., 2002*; *Owji, Nikeghbal & Moosavi, 2018*), and rats have the advantage of larger anatomical structures and a higher blood volume than mice, yielding larger amounts of serum after exsanguination, making easier to perform a

Cienfuegos-Pecina et al. (2020), *PeerJ*, DOI 10.7717/peerj.9438

**Table 3  Biomarkers of kidney and liver injury, oxidative stress biomarkers, and proinflammatory cytokines after administration of succinic acid.**

| Biomarker | SH | 12.5Tox | 25Tox | 50Tox | IR | 12.5+IR | 25+IR | 50+IR |
|---|---|---|---|---|---|---|---|---|
| ALB (g/dL) | 3.1 ± 0.1 | 3.1 ± 0.2 | 3.1 ± 0.2 | 3.3 ± 0.1 | 3.1 ± 0.2 | 3.0 ± 0.2 | 3.0 ± 0.2 | 3.0 ± 0.1 |
| GLU (mg/dL) | 153 ± 15 | 128 ± 57 | 149 ± 29 | 143 ± 22 | 92 ± 15[*] | 103 ± 15 | 101 ± 30 | 92 ± 44 |
| TP (g/dL) | 5.5 ± 0.1 | 5.2 ± 0.5 | 5.3 ± 0.3 | 5.9 ± 0.3 | 5.2 ± 0.4 | 5.2 ± 0.2 | 5.3 ± 0.3 | 5.2 ± 0.1 |
| UA (mg/dL) | 0.7 ± 0.2 | 0.8 ± 0.2 | 0.5 ± 0.1 | 0.6 ± 0.2 | 1.1 ± 0.6 | 0.5 ± 0.1 | 0.6 ± 0.2 | 0.8 ± 0.6 |
| SOD (%) | 94 ± 2 | 88 ± 4 | 91 ± 3 | 90 ± 1 | 97 ± 2 | 92 ± 1 | 88 ± 10 | 91 ± 4 |
| MDA (μmol/g) | 72.71 ± 9.71 | 88.80 ± 14.81 | 85.13 ± 9.87 | 92.10 ± 16.63 | 145.54 ± 97.83 | 71.89 ± 16.91 | 71.64 ± 18.49 | 102.46 ± 24.30 |
| IL-1β (ng/mL) | 1.32 ± 0.70 | 2.28 ± 1.05 | 2.46 ± 0.29 | 4.06 ± 3.08 | 1.50 ± 0.42 | 1.19 ± 0.43 | 1.08 ± 0.48 | 0.83 ± 0.42 |
| IL-6 (ng/mL) | 3.31 ± 1.37 | 3.06 ± 1.01 | 3.08 ± 0.86 | 4.15 ± 0.89 | 3.93 ± 0.72 | 2.26 ± 0.88 | 2.15 ± 0.85 | 1.95 ± 0.92 |
| TNF-α (ng/mL) | 0.77 ± 0.46 | 0.52 ± 0.19 | 0.70 ± 0.17 | 1.24 ± 0.63 | 0.92 ± 0.54 | 0.37 ± 0.18 | 0.30 ± 0.17 | 0.89 ± 1.36 |

Notes.

One-way ANOVA test, Tukey post hoc test. The comparisons were between the 12.5Tox, 25Tox, and SH groups, and between the SH, 12.5+IR, 25+IR, and IR groups.

ALB, albumin; ALP, alkaline phosphatase; GLU, glucose; IL-1β, interleukin 1β; IL-6, interleukin 6; LDH, lactate dehydrogenase; MDA, malondialdehyde; SOD, superoxide dismutase; TNF-α, tumor necrosis factor α; TP, total proteins; UA, uric acid.

[*]$p = 0.0293$ vs. SH.

Cienfuegos-Pecina et al. (2020), *PeerJ*, DOI 10.7717/peerj.9438

**Table 4  Evaluation of renal histopathology after administration of succinic acid.**

| Parameter | SH | 12.5Tox | 25Tox | 50Tox | IR | 12.5+IR | 25+IR | 50+IR |
|---|---|---|---|---|---|---|---|---|
| Tubular Necrosis | 0.00 (0.00–0.00) | 0.00 (0.00–0.75) | 0.00 (0.00–0.00) | 0.00 (0.00–0.75) | 3.50 (3.00-4.00)[*] | 3.00 (2.50–3.50) | 3.50 (3.00–4.00) | 4.00 (3.75-4.00) |
| Acidophilic Casts | 0.00 (0.00–0.50) | 0.00 (0.00–0.00) | 0.00 (0.00–0.75) | 0.00 (0.00–0.00) | 2.00 (2.00–3.00) | 2.00 (1.50–3.00) | 3.00 (2.75–3.25) | 2.00 (2.00-2.50) |
| Vascular Congestion | 1.00 (0.75–2.00) | 1.00 (0.25–1.75) | 0.50 (0.00–1.00) | 1.00 (0.25–1.75) | 1.50 (0.00–3.00) | 2.00 (1.00–2.00) | 1.50 (0.75–3.25) | 3.00 (1.50-4.00) |

**Notes.**

Kruskal-Wallis test, Dunn post hoc test. Values expressed as median (interquartile range). The comparisons were between the 12.5Tox, 25Tox, and SH groups, and between the SH, 12.5+IR, 25+IR, and IR groups. Significant differences when comparing IR vs. SH group: * $p = 0.0297$.

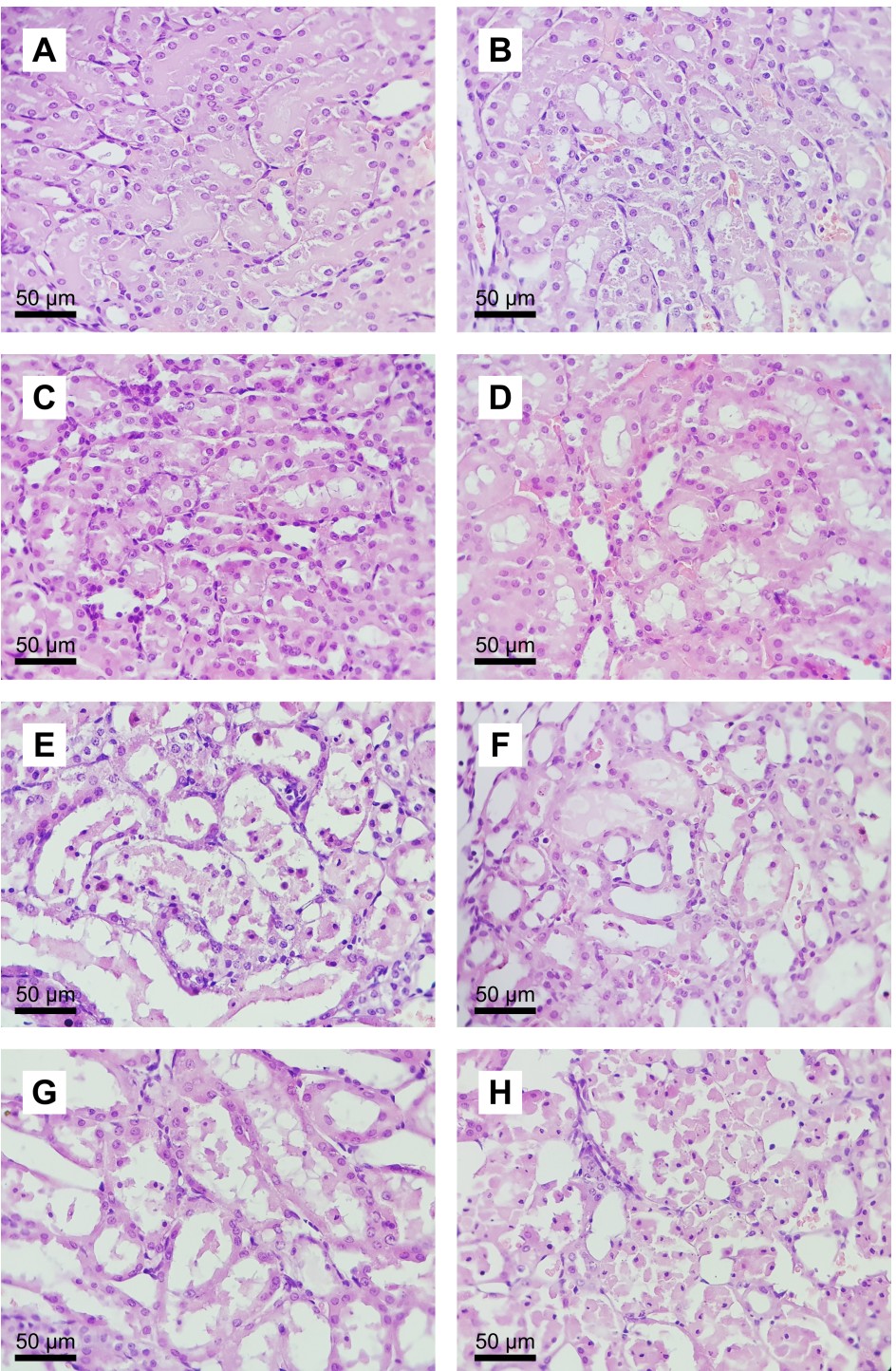

**Figure 8  Microphotographs of the renal histology of the succinic acid-treated groups.** Hematoxylin and eosin staining (original magnification ×400). Conserved tissue architecture was observed in (A) the SH group, (B) the 12.5Tox group, (C) the 25Tox group, and (D) the 50Tox group. Diffuse tubular necrosis was observed in (E) the IR group. Focal tubular necrosis was observed in (F) the 12.5+IR group. Moderate to diffuse tubular necrosis was observed in (G) the 25+IR group. Diffuse and severe tubular necrosis and loss of the microscopic architecture were observed in (H) the 50+IR group.

diversity of biochemical assays. Hence, we decided to use a model of renal IR in rats to assess whether (*S*)-2HG and SA exert a nephroprotective effect.

The length of the periods of ischemia and reperfusion is a major variable to consider in renal IR models. Several studies have reported a broad range of IR conditions. Periods of ischemia of 45 min (*Kang et al., 2005*) and 60 min (*Nakagawa et al., 2005*) have been widely reported in the literature. We decided to use a protocol of 45 min of ischemia and 15 h of reperfusion based on the results of previous studies in our laboratory, which have shown that this protocol causes a significative injury, but is mild enough to identify compounds with moderate nephroprotective activity, allowing us to study deeply their potential pharmacologic activity (*Cura-Esquivel et al., 2018*; *Perez-Meseguer et al., 2019*; *Torres-González et al., 2018*).

It has been widely demonstrated that susceptibility to IR injury is gender-dependent. Studies both in mice and rats have shown that the estradiol-androgen ratio has a key role in the modulation of IR injury. High concentrations of testosterone are associated with a more aggressive injury and higher mortality (*Robert et al., 2011*), and orchiectomy has been shown to attenuate the post-ischemic oxidative stress and the IR injury in mice (*Kim et al., 2006*). We decided to use male and female rats because of the availability of animals in our laboratory. For that reason, the use of independent SH and IR control groups was imperative. The fact that we did not assess each compound both in male and female rats is a limitation of this study. Additional experiments are needed to demonstrate the effect of these compounds on both genders.

Metabolism of (*S*)-2HG has not been well characterized in animal models. Biochemical studies have reported an (*S*)-2HG dehydrogenase activity of $4.5 \pm 5.5$ nmol/min/mg of protein in rat liver, significantly higher than the activity in the kidney (less than 1 nmol/min/mg of protein) and other organs (*Jansen & Wanders, 1993*), but pharmacokinetic experiments are still needed. Studies assessing the chronic exposure to (*S*)-2HG have not been performed, however, the chronic exposure to the (*R*)-2HG enantiomer has already been reported. A study showed that the daily injection of a dose of 250 mg/kg for 32 days caused significant skeletal atrophy and a decrease of the body weight in mice (*Karlstaedt et al., 2016*). Nevertheless, these results cannot be extrapolated to our model, because of the different effects produced by each of the enantiomers of 2-HG in vitro (*Koivunen et al., 2012*). Besides, the application of EGLN inhibitors in the field of solid organ transplantation does not require the use of a chronic exposition, but just an acute administration of the drug. In our study, (*S*)-2HG caused no toxic effects at the hepatic or renal level at the evaluated doses in our experimental conditions, as shown by the normal values of the biochemical markers. Comparison between the nontoxicity groups and the SH group showed no effects of (*S*)-2HG on the biomarkers of oxidative stress, proinflammatory cytokines, and histological parameters. By contrast, (*S*)-2HG exerted a dose-dependent nephroprotective effect in the groups that underwent IR injury, as manifested by the significant amelioration of the changes in the serum concentrations of BUN and creatinine after IR injury. IR injury caused an increase of the serum ALP, AST, and LDH activities, whereas (*S*)-2HG caused a significant amelioration of the activity of these enzymes, which suggests a protective effect of this compound. The TP concentrations differed significantly

between the IR and 25+IR groups, although these values were within the reported reference interval for this rat strain (*Boehm et al., 2007*).

Oxidative stress is one of the main mechanisms involved in IR injury. However, we observed no changes in the biomarkers of oxidative stress after IR injury in our rat model. The magnitude of MDA production and the decrease in tissue SOD activity after IR injury depends on the duration of ischemia and reperfusion. Our results are consistent with those reported previously showing that these changes are nonsignificant after 30 min of ischemia and 24 h of reperfusion (*Dobashi et al., 2000*).

The role of the inflammatory response in the renal IR mechanism is well known and involves the promotion of IL-1 production, which stimulates tubular cells to produce IL-6 and TNF-α (*Daha & Van Kooten, 2000*). However, reports are inconsistent. One study noted that IR does not increase the concentrations of these cytokines (*Zhu et al., 2014*), whereas other studies have reported increases in cytokine concentrations in proportion to the severity of injury (*Shen et al., 2018*; *Tang et al., 2017*; *Zhang et al., 2018*; *Zhang et al., 2015*). In this work, we did not observe significant increases in the concentrations of IL-1β, IL-6, or TNF-α. We also found that (*S*)-2HG did not decrease the concentrations of these proinflammatory cytokines compared with their baseline levels, which suggests that the compound has no immunomodulatory effect. HIF-1α promoted the expression of proinflammatory cytokines such as IL-1α (*Rider et al., 2012*) and IL-1β (*Ogryzko et al., 2019*; *Zhang et al., 2006*) in several experimental models. Because these cytokines stimulate the production of IL-6 and TNF-α, it is possible that this effect antagonizes the potential amelioration of the inflammatory response induced by the nephroprotective effect of (*S*)-2HG. Additional studies are required to confirm this idea.

Renal IR caused significant injury to tissue architecture, which appears as acute tubular necrosis. We observed a tendency for (*S*)-2HG toward amelioration of tissue injury in the three histological parameters evaluated, but these changes were not statistically significant. These results agree with those of other studies of compounds with nephroprotective activity, in which the magnitude of tissue injury was assessed using a semiquantitative approach (*Kobuchi et al., 2009*; *Torres-González et al., 2018*). The use of morphometric analysis would have allowed us to quantify the ratio of damaged to undamaged renal tubules and to use parametric statistic tests, as reported previously (*Barrera-Chimal et al., 2011*), to examine this question at the tissue level.

(*S*)-2HG administration did not exert an unequivocal accumulation of HIF-1α in the experimental conditions used in our study, as it was shown by the measurements performed in kidney tissue by ELISA and Western blot. It has been shown that the administration of EGLN inhibitors 6–8 h before ischemia causes HIF-1α accumulation (*Bernhardt et al., 2009*; *Xu et al., 2011*). In the present study, IR injury was induced 8 h after the final dose of (*S*)-2HG and tissue HIF-1α concentration was measured 15 h after IR injury. HIF-1α is a highly regulated protein, extremely sensitive to the concentration of $O_2$ in the microenvironment and easily degraded (in this study we observed a semi-degraded 55 kDa isoform of HIF-1α in the Western blot, as referred by several manufacturers of anti-HIF-1α antibodies). For that reason, it is understandable that the expression of this protein would not be stable after the long reperfusion period; however, the IR group showed a significant
accumulation of HIF-1α after the 15-hours reperfusion period, but the 12.5+IR and 25+IR groups did not follow this behavior. The effect observed in the IR group is consistent with that of a study in which HIF-1α accumulation caused by IR occurred in two phases, namely an acute phase during ischemia and a late phase during reperfusion (*Conde et al., 2012*). The findings of the present study suggest that the activation of a nephroprotective mechanism by the treatment with (*S*)-2HG could be related whit the non-accumulation of HIF-1α during the late reperfusion period in the treated groups. Differences in the patterns of HIF-1α expression measured by ELISA or Western blot could be explained based on the instability of HIF-1α. Different sample treatments resulted in distinct grades of degradation, requiring additional experiments to comprehensively assess the effect of sample preparation in the HIF-1α concentrations.

Despite the inherent instability of HIF-1α in these experimental conditions, the expression of genes directly transcribed by HIF-1 is an indirect way to infer a previous stabilization of HIF-1α caused by the administration of (*S*)-2HG. Inducible heme oxygenase (heme oxygenase 1, Hmox1) is a microsomal membrane enzyme, which catalyzes the oxidative cleavage of heme molecules, yielding biliverdin, CO and iron. The expression of this enzyme increases significatively in response to hypoxia (it is hypothesized that biliverdin exerts an antioxidant effect, antagonistic to the heme-mediated production of ROS), and it is regulated directly by HIF-1 (*Lee et al., 1997*). It has been described that the inhibition of the EGLN family induces a 4–5 fold increase in the Hmox1 expression in a model of kidney transplantation in rats and that the expression of this gene is stable for at least 24 h after the inhibition of the EGLN family (*Bernhardt et al., 2009*). In this study, we observed a 14-fold increase in the expression of Hmox1 in the 25Tox group compared to the SH group. This significant increase in the expression of Hmox1 strongly suggests a role of the HIF-1 pathway in the nephroprotective effect showed by the (*S*)-2HG. It has been described that the IR injury induces *per se* the expression of Hmox1 in rats and that the transcript is also stable for at least 24 h (*Maines et al., 1993*). In this study, we observed that IR injury drastically increased the expression of Hmox1 in the IR, 12.5+IR, and 25+IR groups, with a tendency to increase its expression in the 25+IR group. Additional studies are needed to fully characterize the biochemical pathways modified by (*S*)-2HG in the IR injury, in the kidney, and in other organs.

By contrast, we observed no toxic effects of SA at the hepatic or renal level at the doses used, as shown by the normal biomarker values, which agrees with the findings of *Maekawa et al. (1990)* who reported the absence of a toxic effect in rats of the F344 strain. In our study, it was not observed an alteration in the proinflammatory cytokines, the oxidative stress biomarkers, or the histological parameters.

Unlike (*S*)-2HG, which showed a significant nephroprotective effect, SA tended to aggravate IR injury, as shown by the inability to ameliorate the changes in BUN and creatinine concentrations at most of the doses used, although there was a trend toward a dose-dependent effect. We found no changes in the other biomarkers evaluated after SA administration. The role of SA in the IR injury processes has been extensively investigated. SA accumulates significantly during the IR injury processes, such as in brain stroke, and its intracellular concentration can increase by up to 30-fold (*Chouchani et al., 2014*;

*Sahni et al., 2017*). SA plays a role in the reverse electron transport (RET) phenomenon (*Scialò, Fernández-Ayala & Sanz, 2017*), which occurs in the inner mitochondrial membrane when complexes III and IV of the electron transport chain are saturated and RET occurs in complex I and ROS are generated (*Chouchani et al., 2014*). Stimulation of complex II (succinate dehydrogenase) strongly promotes the RET phenomenon (*Chance & Hollunger, 1961*), and it is probable that SA administration before IR induction activates this mechanism and produces large amounts of ROS. Additionally, an in vitro study has demonstrated that, despite its EGLN-1 inhibitory activity, SA cannot cause intracellular HIF-1α accumulation (*Koivunen et al., 2007*).

SA did not affect the biomarkers of oxidative stress or proinflammatory cytokines. Additional studies under different IR conditions are needed to examine the behavior of these mediators in response to this compound. Histological analysis of the SA-treated groups showed a dose-dependent tendency toward increased histological evidence of injury. Although this response was not statistically significant, it is possible that the effect was underestimated by the nonparametric test used to analyze the semiquantitative data. Further quantitative morphometric studies are needed to confirm whether SA affects tissue architecture in association with IR injury.

As previously described, inhibitors of the EGLN family, such as dimethyloxalylglycine and L-mimosine, promote tissue HIF-1α accumulation and significantly ameliorate the changes in serum BUN and creatinine levels and histological damage after IR injury (*Hill et al., 2008*). Similarly, the administration of the EGLN inhibitor FG-4497 has been shown to improve graft survival in an allogenic kidney transplantation model (*Bernhardt et al., 2009*). The results of our study suggest that (*S*)-2HG is a potential candidate for additional studies in animal models. The effects caused by SA administration are similar to those reported for mitochondrial ROS production by RET. Therefore, subsequent evaluation of this compound in other IR models may help to elucidate the mechanisms underlying these metabolic pathways, as well as the possible clinical implications.

## CONCLUSIONS

Neither (*S*)-2HG nor SA showed a hepato- or nephrotoxic effect at the doses tested. (*S*)-2HG showed a dose-dependent nephroprotective effect against IR injury, which involved amelioration of kidney injury biomarkers and an increase in the expression of Hmox1, suggesting stabilization of HIF-1α. SA did not show a nephroprotective effect but tended to increase IR injury when given at high doses. Neither (*S*)-2HG nor SA exhibited immunomodulatory or antioxidant activity at the different doses used here.

### Funding

This study was financially supported by resources of the Liver Unit and the Department of Analytical Chemistry and did not receive any specific grant from funding agencies in the public, commercial, or not-for-profit sectors. Eduardo Cienfuegos-Pecina received a

scholarship from the National Council of Science and Technology (CONACYT) through the project 2017-01-5652. There was no additional external funding received for this study. The funders had no role in study design, data collection and analysis, decision to publish, or preparation of the manuscript.

### Grant Disclosures
The following grant information was disclosed by the authors:
Liver Unit and the Department of Analytical Chemistry.
National Council of Science and Technology (CONACYT): 2017-01-5652.

### Competing Interests
The authors declare there are no competing interests.

### Author Contributions

- Eduardo Cienfuegos-Pecina conceived and designed the experiments, performed the experiments, analyzed the data, prepared figures and/or tables, authored or reviewed drafts of the paper, and approved the final draft.
- Tannya R. Ibarra-Rivera, Alma L. Saucedo, Gabriela Alarcon-Galvan, Diana Raquel Rodríguez-Rodríguez and Paula Cordero-Pérez conceived and designed the experiments, performed the experiments, analyzed the data, prepared figures and/or tables, authored or reviewed drafts of the paper, contributed reagents, materials, analysis tools, and approved the final draft.
- Luis A. Ramírez-Martínez, Deanna Esquivel-Figueroa, Ixel Domínguez-Vázquez and Karina J. Alcántara-Solano performed the experiments, analyzed the data, authored or reviewed drafts of the paper, and approved the final draft.
- Diana P. Moreno-Peña conceived and designed the experiments, performed the experiments, analyzed the data, authored or reviewed drafts of the paper, and approved the final draft.
- Liliana Torres-González conceived and designed the experiments, performed the experiments, analyzed the data, prepared figures and/or tables, authored or reviewed drafts of the paper, and approved the final draft.
- Linda E. Muñoz-Espinosa and Edelmiro Pérez-Rodríguez analyzed the data, authored or reviewed drafts of the paper, contributed reagents, materials, analysis tools, and approved the final draft.

### Animal Ethics
The following information was supplied relating to ethical approvals (i.e., approving body and any reference numbers):

The Ethics and Research Committee of the School of Medicine, Universidad Autónoma de Nuevo León provided full approval for this research (register number HI17-0002).

### Data Availability
The raw measurements are available in the Supplemental Files.

## Supplemental Information

Supplemental information for this article can be found online at http://dx.doi.org/10.7717/peerj.9438#supplemental-information.

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

# PeerJ

**Cai Z, Zhong H, Bosch-Marce M, Fox-Talbot K, Wang L, Wei C, Trush MA, Semenza GL. 2008.** Complete loss of ischaemic preconditioning-induced cardioprotection in mice with partial deficiency of HIF-1α. *Cardiovascular Research* **77**:463–470 DOI 10.1093/cvr/cvm035.

**CENATRA. 2018.** *Reporte Anual 2018 de Donación y Trasplantes en México.* México: Centro Nacional de Trasplantes.

**Chance B, Hollunger G. 1961.** The interaction of energy and electron transfer reactions in mitochondria: I. General properties and nature of the products of succinate-linked reduction of pyridine nucleotide. *Journal of Biological Chemistry* **236**:1534–1543.

**Chen F, Date H. 2015.** Update on ischemia-reperfusion injury in lung transplantation. *Current Opinion in Organ Transplantation* **20**:515–520 DOI 10.1097/MOT.0000000000000234.

**Chouchani ET, Pell VR, Gaude E, Aksentijević D, Sundier SY, Robb EL, Logan A, Nadtochiy SM, Ord ENJ, Smith AC, Eyassu F, Shirley R, Hu C-H, Dare AJ, James AM, Rogatti S, Hartley RC, Eaton S, Costa ASH, Brookes PS, Davidson SM, Duchen MR, Saeb-Parsy K, Shattock MJ, Robinson AJ, Work LM, Frezza C, Krieg T, Murphy MP. 2014.** Ischaemic accumulation of succinate controls reperfusion injury through mitochondrial ROS. *Nature* **515**:431–435 DOI 10.1038/nature13909.

**Conde E, Alegre L, Blanco-Sánchez I, Sáenz-Morales D, Aguado-Fraile E, Ponte B, Ramos E, Sáiz A, Jiménez C, Ordoñez A, López-Cabrera M, del Peso L, de Landázuri MO, Liaño F, Selgas R, Sanchez-Tomero JA, García-Bermejo ML. 2012.** Hypoxia inducible factor 1-alpha (HIF-1 alpha) is induced during reperfusion after renal ischemia and is critical for proximal tubule cell survival. *PLOS ONE* **7**:e33258–e33258 DOI 10.1371/journal.pone.0033258.

**Cura-Esquivel I, Delgado-Chávez EN, García-Narro JH, Torres-González L, Alarcón-Galván G, Moreno-Peña DP, Esquivel-Figueroa D, Cantú-Machuca DV, Muñoz Espinosa LE, Garza-Ocañas L, Cordero-Pérez P. 2018.** Attenuation of pro-inflammatory cytokines and oxidative stress by misoprostol in renal ischemia/reperfusion in rats. *Die Pharmazie - An International Journal of Pharmaceutical Sciences* **73**:537–540 DOI 10.1691/ph.2018/8498.

**Daha MR, Van Kooten C. 2000.** Is the proximal tubular cell a proinflammatory cell? *Nephrology Dialysis Transplantation* **15**:41–43 DOI 10.1093/ndt/15.suppl_6.41.

**Dang L, White DW, Gross S, Bennett BD, Bittinger MA, Driggers EM, Fantin VR, Jang HG, Jin S, Keenan MC, Marks KM, Prins RM, Ward PS, Yen KE, Liau LM, Rabinowitz JD, Cantley LC, Thompson CB, Vander Heiden MG, Su SM. 2009.** Cancer-associated IDH1 mutations produce 2-hydroxyglutarate. *Nature* **462**:739–744 DOI 10.1038/nature08617.

**Dengler VL, Galbraith MD, Espinosa JM. 2014.** Transcriptional regulation by hypoxia inducible factors. *Critical Reviews in Biochemistry and Molecular Biology* **49**:1–15 DOI 10.3109/10409238.2013.838205.

**Dobashi K, Ghosh B, Orak JK, Singh I, Singh AK. 2000.** Kidney ischemia-reperfusion: modulation of antioxidant defenses. *Molecular and Cellular Biochemistry* **205**:1–11 DOI 10.1023/a:1007047505107.

**Dorweiler B, Pruefer D, Andrasi TB, Maksan SM, Schmiedt W, Neufang A, Vahl CF. 2007.** Ischemia-reperfusion injury. *European Journal of Trauma and Emergency Surgery* **33**:600–612 DOI 10.1007/s00068-007-7152-z.

**Granger DN. 1988.** Role of xanthine oxidase and granulocytes in ischemia-reperfusion injury. *American Journal of Physiology-Heart and Circulatory Physiology* **255**:H1269–H1275 DOI 10.1152/ajpheart.1988.255.6.H1269.

**Hausenloy DJ, Yellon DM. 2007.** Reperfusion injury salvage kinase signalling: taking a RISK for cardioprotection. *Heart Failure Reviews* **12**:217–234 DOI 10.1007/s10741-007-9026-1.

**Heyman SN, Lieberthal W, Rogiers P, Bonventre JV. 2002.** Animal models of acute tubular necrosis. *Current Opinion in Critical Care* **8**:526–534 DOI 10.1097/00075198-200212000-00008.

**Hill P, Shukla D, Tran MGB, Aragones J, Cook HT, Carmeliet P, Maxwell PH. 2008.** Inhibition of hypoxia inducible factor hydroxylases protects against renal ischemia-reperfusion injury. *Journal of the American Society of Nephrology* **19**:39–46 DOI 10.1681/asn.2006090998.

**Ivan M, Kondo K, Yang H, Kim W, Valiando J, Ohh M, Salic A, Asara JM, Lane WS, Kaelin Jr WG. 2001.** HIFα targeted for VHL-mediated destruction by proline hydroxylation: implications for O2 sensing. *Science* **292**:464–468 DOI 10.1126/science.1059817.

**Jaakkola P, Mole DR, Tian Y-M, Wilson MI, Gielbert J, Gaskell SJ, Kriegsheim Av, Hebestreit HF, Mukherji M, Schofield CJ, Maxwell PH, Pugh CW, Ratcliffe PJ. 2001.** Targeting of HIF-α to the von Hippel-Lindau Ubiquitylation Complex by O2-regulated prolyl hydroxylation. *Science* **292**:468–472 DOI 10.1126/science.1059796.

**Jansen GA, Wanders RJA. 1993.** l-2-Hydroxyglutarate dehydrogenase: identification of a novel enzyme activity in rat and human liver. Implications for l-2-hydroxyglutaric acidemia. *Biochimica et Biophysica Acta (BBA) - Molecular Basis of Disease* **1225**:53–56 DOI 10.1016/0925-4439(93)90121-G.

**Kaelin WG, Ratcliffe PJ. 2008.** Oxygen sensing by metazoans: the central role of the HIF hydroxylase pathway. *Molecular Cell* **30**:393–402 DOI 10.1016/j.molcel.2008.04.009.

**Kalakech H, Tamareille S, Pons S, Godin-Ribuot D, Carmeliet P, Furber A, Martin V, Berdeaux A, Ghaleh B, Prunier F. 2013.** Role of hypoxia inducible factor-1 α in remote limb ischemic preconditioning. *Journal of Molecular and Cellular Cardiology* **65**:98–104 DOI 10.1016/j.yjmcc.2013.10.001.

**Kang DG, Sohn EJ, Moon MK, Lee YM, Lee HS. 2005.** Rehmannia glutinose ameliorates renal function in the Ischemia/Reperfusion-induced acute renal failure rats. *Biological and Pharmaceutical Bulletin* **28**:1662–1667 DOI 10.1248/bpb.28.1662.

**Karlstaedt A, Zhang X, Vitrac H, Harmancey R, Vasquez H, Wang JH, Goodell MA, Taegtmeyer H. 2016.** Oncometabolite D-2-hydroxyglutarate impairs α-ketoglutarate dehydrogenase and contractile function in rodent heart. *Proceedings of the National Academy of Sciences of the United States of America* **113**:10436–10441 DOI 10.1073/pnas.1601650113.

**Karuppagounder SS, Ratan RR. 2012.** Hypoxia-inducible factor prolyl hydroxylase inhibition: robust new target or another big bust for stroke therapeutics? *Journal of Cerebral Blood Flow & Metabolism* **32**:1347–1361 DOI 10.1038/jcbfm.2012.28.

**Kim J, Kil IS, Seok YM, Yang ES, Kim DK, Lim DG, Park J-W, Bonventre JV, Park KM. 2006.** Orchiectomy attenuates post-ischemic oxidative stress and ischemia/reperfusion injury in mice: a role for manganese superoxide dismutase. *Journal of Biological Chemistry* **281**:20349–20356 DOI 10.1074/jbc.M512740200.

**Kobuchi S, Shintani T, Sugiura T, Tanaka R, Suzuki R, Tsutsui H, Fujii T, Ohkita M, Ayajiki K, Matsumura Y. 2009.** Renoprotective effects of gamma-aminobutyric acid on ischemia/reperfusion-induced renal injury in rats. *European Journal of Pharmacology* **623**:113–118 DOI 10.1016/j.ejphar.2009.09.023.

**Koivunen P, Hirsilä M, Remes AM, Hassinen IE, Kivirikko KI, Myllyharju J. 2007.** Inhibition of Hypoxia-inducible Factor (HIF) hydroxylases by citric acid cycle intermediates: possible links between cell metabolism and stabilization of HIF. *Journal of Biological Chemistry* **282**:4524–4532 DOI 10.1074/jbc.M610415200.

**Koivunen P, Lee S, Duncan CG, Lopez G, Lu G, Ramkissoon S, Losman JA, Joensuu P, Bergmann U, Gross S, Travins J, Weiss S, Looper R, Ligon KL, Verhaak RGW, Yan H, Kaelin Jr WG. 2012.** Transformation by the (R)-enantiomer of 2-hydroxyglutarate linked to EGLN activation. *Nature* **483**:484–488 DOI 10.1038/nature10898.

**Kolitz M, Cohen-Arazi N, Hagag I, Katzhendler J, Domb AJ. 2009.** Biodegradable polyesters derived from amino acids. *Macromolecules* **42**:4520–4530 DOI 10.1021/ma900464g.

**Lee PJ, Jiang B-H, Chin BY, Iyer NV, Alam J, Semenza GL, Choi AMK. 1997.** Hypoxia-inducible Factor-1 mediates transcriptional activation of the heme oxygenase-1 gene in response to hypoxia. *Journal of Biological Chemistry* **272**:5375–5381 DOI 10.1074/jbc.272.9.5375.

**Li YW, Zhang Y, Zhang L, Li X, Yu JB, Zhang HT, Tan BB, Jiang LH, Wang YX, Liang Y, Zhang XS, Wang WS, Liu HG. 2014.** Protective effect of tea polyphenols on renal ischemia/reperfusion injury via suppressing the activation of TLR4/NF-kappaB p65 signal pathway. *Gene* **542**:46–51 DOI 10.1016/j.gene.2014.03.021.

**Maekawa A, Todate A, Onodera H, Matsushima Y, Nagaoka T, Shibutani M, Ogasawara H, Kodama Y, Hayashi Y. 1990.** Lack of toxicity/carcinogenicity of monosodium succinate in F344 rats. *Food and Chemical Toxicology* **28**:235–241 DOI 10.1016/0278-6915(90)90035-L.

**Maines MD, Mayer RD, Ewing JF, McCoubrey WK. 1993.** Induction of kidney heme oxygenase-1 (HSP32) mRNA and protein by ischemia/reperfusion: possible role of heme as both promotor of tissue damage and regulator of HSP32. *Journal of Pharmacology and Experimental Therapeutics* **264**:457–462.

**Markgraf JH, Davis HA. 1990.** Steric course of lactonization in the deamination of glutamic acid: an organic mechanism experiment. *Journal of Chemical Education* **67**:173–174 DOI 10.1021/ed067p173.

**McManaman JL, Bain DL. 2002.** Structural and conformational analysis of the oxidase to dehydrogenase conversion of xanthine oxidoreductase. *Journal of Biological Chemistry* **277**:21261–21268 DOI 10.1074/jbc.M200828200.

**Milkiewicz M, Pugh CW, Egginton S. 2004.** Inhibition of endogenous HIF inactivation induces angiogenesis in ischaemic skeletal muscles of mice. *The Journal of Physiology* **560**:21–26 DOI 10.1113/jphysiol.2004.069757.

**Murry CE, Jennings RB, Reimer KA. 1986.** Preconditioning with ischemia: a delay of lethal cell injury in ischemic myocardium. *Circulation* **74**:1124–1136 DOI 10.1161/01.CIR.74.5.1124.

**Myllyharju J. 2009.** HIF prolyl 4-hydroxylases and their potential as drug targets. *Current Pharmaceutical Design* **15**:3878–3885 DOI 10.2174/138161209789649457.

**Nakagawa T, Yokozawa T, Satoh A, Kim HY. 2005.** Attenuation of renal ischemia-reperfusion injury by proanthocyanidin-rich extract from grape seeds. *Journal of Nutritional Science and Vitaminology* **51**:283–286 DOI 10.3177/jnsv.51.283.

**Neuberger A. 1948.** Stereochemistry of amino acids. *Advances in Protein Chemistry* **4**:297–383 DOI 10.1016/S0065-3233(08)60009-1.

**Ogryzko NV, Lewis A, Wilson HL, Meijer AH, Renshaw SA, Elks PM. 2019.** HIF-1α–induced expression of Il-1β protects against mycobacterial infection in zebrafish. *The Journal of Immunology* **202**:494–502 DOI 10.4049/jimmunol.1801139.

**Ohh M, Park CW, Ivan M, Hoffman MA, Kim T-Y, Huang LE, Pavletich N, Chau V, Kaelin WG. 2000.** Ubiquitination of hypoxia-inducible factor requires direct binding to the β-domain of the von Hippel–Lindau protein. *Nature Cell Biology* **2**:423–427 DOI 10.1038/35017054.

**Owji SM, Nikeghbal E, Moosavi SM. 2018.** Comparison of ischaemia–reperfusion-induced acute kidney injury by clamping renal arteries, veins or pedicles in anaesthetized rats. *Experimental Physiology* **103**:1390–1402 DOI 10.1113/ep087140.

**Pauli GF, Jaki BU, Lankin DC. 2007.** A routine experimental protocol for qHNMR illustrated with taxol. *Journal of Natural Products* **70**:589–595 DOI 10.1021/np060535r.

**Perez-Meseguer J, Torres-González L, Gutiérrez-González JA, Alarcón-Galván G, Zapata-Chavira H, Waksman-de Torres N, Moreno-Peña DP, Muñoz-Espinosa LE, Cordero-Pérez P. 2019.** Anti-inflammatory and nephroprotective activity of Juglans mollis against renal ischemia–reperfusion damage in a Wistar rat model. *BMC Complementary and Alternative Medicine* **19**:186 DOI 10.1186/s12906-019-2604-7.

**Pleissner D, Wimmer R, Eriksen NT. 2011.** Quantification of amino acids in fermentation media by isocratic HPLC analysis of their α-hydroxy acid derivatives. *Analytical Chemistry* **83**:175–181 DOI 10.1021/ac1021908.

**Przyklenk K, Bauer B, Ovize M, Kloner RA, Whittaker P. 1993.** Regional ischemic 'preconditioning' protects remote virgin myocardium from subsequent sustained coronary occlusion. *Circulation* **87**:893–899 DOI 10.1161/01.CIR.87.3.893.

**Randhawa PK, Bali A, Jaggi AS. 2015.** RIPC for multiorgan salvage in clinical settings: evolution of concept, evidences and mechanisms. *European Journal of Pharmacology* **746**:317–332 DOI 10.1016/j.ejphar.2014.08.016.

**Rider P, Kaplanov I, Romzova M, Bernardis L, Braiman A, Voronov E, Apte R. 2012.** The transcription of the alarmin cytokine interleukin-1 alpha is controlled by hypoxia inducible factors 1 and 2 alpha in hypoxic cells. *Frontiers in Immunology* **3**:290 DOI 10.3389/fimmu.2012.00290.

**Ritthausen H. 1872.** Ueber das Drehungsvermögen von Glutan- und Aepfelsäure. *Journal für Praktische Chemie* **5**:354–355 DOI 10.1002/prac.18720050132.

**Robert R, Ghazali DA, Favreau F, Mauco G, Hauet T, Goujon J-M. 2011.** Gender difference and sex hormone production in rodent renal ischemia reperfusion injury and repair. *Journal of Inflammation* **8**:14 DOI 10.1186/1476-9255-8-14.

**Sahni PV, Zhang J, Sosunov S, Galkin A, Niatsetskaya Z, Starkov A, Brookes PS, Ten VS. 2017.** Krebs cycle metabolites and preferential succinate oxidation following neonatal hypoxic-ischemic brain injury in mice. *Pediatric Research* **83**:491–497 DOI 10.1038/pr.2017.277.

**Salvadori M, Rosso G, Bertoni E. 2015.** Update on ischemia-reperfusion injury in kidney transplantation: pathogenesis and treatment. *World journal of transplantation* **5**:52–67 DOI 10.5500/wjt.v5.i2.52.

**Scialò F, Fernández-Ayala DJ, Sanz A. 2017.** Role of mitochondrial reverse electron transport in ROS signaling: potential roles in health and disease. *Frontiers in Physiology* **8**:428 DOI 10.3389/fphys.2017.00428.

**Semenza GL. 2014.** Oxygen sensing, hypoxia-inducible factors, and disease pathophysiology. *Annual Review of Pathology: Mechanisms of Disease* **9**:47–71 DOI 10.1146/annurev-pathol-012513-104720.

**Shen Y, Qiu T, Liu X, Zhang L, Wang Z, Zhou JJERMPS. 2018.** Renal ischemia-reperfusion injury attenuated by splenic ischemic preconditioning. *European Review for Medical and Pharmacological Sciences* **22**:2134–2142 DOI 10.26355/eurrev_201804_14747.

**Sigma-Aldrich P. 2018.** L-α-Hydroxyglutaric acid disodium salt. *Available at https: //www.sigmaaldrich.com/catalog/product/sigma/90790?lang=es®ion=AR& gclid=Cj0KCQjw5s3cBRCAARIsAB8ZjU1pofY_n2QisjYfRVrds7Sstahb8DVPPq5K3Y-eXzWGOR9Ki_-8_6IaAn51EALw_wcB* (accessed on 8 September 2018).

**Takasu C, Vaziri ND, Li S, Robles L, Vo K, Takasu M, Pham C, Farzaneh SH, Shimada M, Stamos MJ, Ichii H. 2017.** Treatment with dimethyl fumarate ameliorates liver ischemia/reperfusion injury. *World Journal of Gastroenterology* **23**:4508–4516 DOI 10.3748/wjg.v23.i25.4508.

**Tang CY, Lai CC, Huang PH, Yang AH, Chiang SC, Huang PC, Tseng KW, Huang CH. 2017.** Magnolol reduces renal ischemia and reperfusion injury via inhibition of apoptosis. *American Journal of Chinese Medicine* **45**:1421–1439 DOI 10.1142/s0192415x1750077x.

**Torres-González L, Cienfuegos-Pecina E, Perales-Quintana MM, Alarcon-Galvan G, Muñoz Espinosa LE, Pérez-Rodríguez E, Cordero-Pérez P. 2018.** Nephroprotective effect of sonchus oleraceus extract against kidney injury induced by ischemia-reperfusion in wistar rats. *Oxidative Medicine and Cellular Longevity* **2018**:9572803 DOI 10.1155/2018/9572803.

**Van Draanen NA, Hengst S. 2010.** The conversion of l-Phenylalanine to (S)-2-Hydroxy-3-phenylpropanoic Acid: a simple, visual example of a stereospecific SN2 reaction. *Journal of Chemical Education* **87**:623–624 DOI 10.1021/ed100167k.

**Wang GL, Semenza GL. 1995.** Purification and characterization of hypoxia-inducible factor 1. *Journal of Biological Chemistry* **270**:1230–1237 DOI 10.1074/jbc.270.3.1230.

**Wei Q, Dong Z. 2012.** Mouse model of ischemic acute kidney injury: technical notes and tricks. *American Journal of Physiology - Renal Physiology* **303**:F1487–F1494 DOI 10.1152/ajprenal.00352.2012.

**Welbourn C, Goldman G, Paterson I, Valeri C, Shepro D, Hechtman H. 1991.** Pathophysiology of ischaemia reperfusion injury: central role of the neutrophil. *British Journal of Surgery* **78**:651–655 DOI 10.1002/bjs.1800780607.

**Williams L, Nguyen T, Li Y, Porter TN, Raushel FM. 2006.** Uronate isomerase: a nonhydrolytic member of the amidohydrolase superfamily with an ambivalent requirement for a divalent metal ion. *Biochemistry* **45**:7453–7462 DOI 10.1021/bi060531l.

**Xu W, Yang H, Liu Y, Yang Y, Wang P, Kim S-H, Ito S, Yang C, Wang P, Xiao M-T, Liu L-x, Jiang W-q, Liu J, Zhang J-y, Wang B, Frye S, Zhang Y, Xu Y-h, Lei Q-y, Guan K-L, Zhao S-m, Xiong Y. 2011.** Oncometabolite 2-Hydroxyglutarate is a competitive inhibitor of alpha-ketoglutarate-dependent dioxygenases. *Cancer Cell* **19**:17–30 DOI 10.1016/j.ccr.2010.12.014.

**Yildiz F, Coban S, Terzi A, Aksoy N, Bitiren M. 2015.** Protective effect of micronized purified flavonoid fraction on ischemia/reperfusion injury of rat liver. *Transplantation Proceedings* **47**:1507–1510 DOI 10.1016/j.transproceed.2015.04.062.

**Zhang J, Tang L, Li GS, Wang J. 2018.** The anti-inflammatory effects of curcumin on renal ischemia-reperfusion injury in rats. *Renal Failure* **40**:680–686 DOI 10.1080/0886022X.2018.1544565.

**Zhang T, Ma Y, Xu KQ, Huang WQ. 2015.** Pretreatment of parecoxib attenuates hepatic ischemia/reperfusion injury in rats. *BMC Anesthesiology* **15**:165 DOI 10.1186/s12871-015-0147-0.

**Zhang W, Petrovic J-M, Callaghan D, Jones A, Cui H, Howlett C, Stanimirovic D. 2006.** Evidence that hypoxia-inducible factor-1 (HIF-1) mediates transcriptional activation of interleukin-1β (IL-1β) in astrocyte culturess. *Journal of Neuroimmunology* **174**:63–73 DOI 10.1016/j.jneuroim.2006.01.014.

**Zhu S-h, Zhou L-j, Jiang H, Chen R-j, Lin C, Feng S, Jin J, Chen J-h, Wu J-y. 2014.** Protective effect of indomethacin in renal ischemia-reperfusion injury in mice. *Journal of Zhejiang University Science B* **15**:735–742 DOI 10.1631/jzus.B1300196.

**Zollinger H. 1995.** Dediazoniation Reactions Involving Diazonium Ion Intermediates: Section 7.5–7.8. In: Zollinger H, ed. *Diazo chemistry II: aliphatic, inorganic and organometallic compounds.* Hoboken: Wiley, 241–277.