# Peer review of "Effect of sodium (S)-2-hydroxyglutarate in male, and succinic acid in female Wistar rats against renal ischemia-reperfusion injury, suggesting a role of the HIF-1 pathway"

_PeerJ, doi:10.7717/peerj.9438_

## Round 0.1 · original submission · Major Revisions

Your manuscript has been revised by 3 reviewers and they recommended major review. The question of using male or female rats with different potential protective agents is a critical point and the title of the manuscript has to contain this. Note that your manuscript will need a second round of review. I recommend you to indicate in the rebuttal letter in a point by point where the changes you made appear in the revised text (page, lines, etc).

Reviewer 1 ·

Basic reporting

No comment

Experimental design

1. The best and standard model of renal IR injury must be C57mouse. Why did the authors choose Wister rats?
2. In the reference 32 by Wei and Dong, a mouse model was used, and they chose the back incision, but in the present study, the author chose the midline incision. Was this reference proper?
3. Why did you choose 15 hours reperfusion but not 24 hours, please provide a reasonable explanation?

Validity of the findings

1. The elevation of HIF may be critical for the effectiveness and mechanism of the two drugs. So, the only detection of HIF using ELISA in tissue homogenates. The author should supplement the results of Western Blot.
2. The statements in the Result sections may be improper, such as "There were no
significant differences between A and B." A more explicit statement must like "A is higher than B, or 2-HG decreased the level of creatinine."
3. The nephroprotective effects must be most important, which should be firstly described in the Result section
4. If possible, other interventions should be used as a positive control, such as local IPC or remote IPC.

Additional comments

No comments

·

Basic reporting

No comment.

Experimental design

Please have a look at Major concerns: 1 to 6.

Validity of the findings

No comment.

Additional comments

Kidney transplantation is the definitive treatment for patients with end-stage chronic kidney disease. However, the shortage of available organs for transplantation is a formidable challenge. Ischemia–reperfusion (IR) injury is the main cause of delayed graft function in solid organ transplantation. It is inspiriting and beneficial to analyze the activities and potential pharmacological properties of preconditioning drugs (hypoxia mimetics) on preventing or limitting IR injury. In this manuscript, Dr. Eduardo Cienfuegos-Pecina and colleagues looked at the nephroprotective effect of the egg-laying defective nine homolog (EGLN) inhibitors, (S)-2-hydroxyglutarate [(S)-2HG] and succinic acid (SA) in IR injury rat. The study’s conclusion showed that only (S)-2HG showed a dose-dependent nephroprotective effect at the evaluated doses, which involved the promotion of HIF-1alpha accumulation. Although the current study is well and interesting, there are some major concerns that need to be addressed before consideration of publication.

Major concerns:
1. “Materials & Methods – Animals” Line 158-165. “Wistar rats, weighing 250–300 g, were used: 32 male rats to evaluate (S)-2HG activity and 42 females to evaluate SA activity.” Why study the activity of (S)-2HG in male rats and SA in female rats?
2. “Materials & Methods – Experimental design” Line 168-202. “Rats were treated with (S)-2HG at a dose of 12.5 or 25 mg/kg (12.5Tox and 25Tox, respectively) in double-distilled water administered p.o. twice per day for 2 days.” Please give clear reasons and references for the above concentration and administration time.
3. “Materials & Methods – Experimental design” Line 168-202. “Rats were treated with SA at a dose of 12.5, 25, or 50 mg/kg (12.5Tox, 25Tox, and 50Tox, respectively) in double-distilled water under the same conditions as the SH group.” Please give clear reasons and references for the above concentration and administration time.
4. “Table 2 & Table 4”. The Scores of Tubular Necrosis and Acidophilic Casts in IR group were 4.00 (4.00-4.00) and 4.00 (3.00-4.00) respectively in Table 2. However, the Scores of Tubular Necrosis and Acidophilic Casts in IR group were 3.50 (3.00-4.00) and 2.00 (2.00-3.00) in Table 4. It shows that the Score of Acidophilic Casts was different between the two IR groups. Why is there a difference? Combined with the first major question, is it a gender difference?
5. “Figure 2 & Figure 4”. The serum concentrations of creatinine can reflects kidney function. It seem that the creatinine of IR group was about 3.1 mg/dL in Figure 2. However, the creatinine of IR group was about 2.5 mg/dL in Figure 4. It shows that the creatinine was different between the two IR groups. Why is there a difference? Combined with the major question 1 and 4, is it a gender difference?
6. “Results – Evaluation of the effect of sodium (S)-2HG treatment on the tissue HIF-1α concentration” Line 382-389 and Figure 6. Although the results of the manuscript showed that there was not remarkable nephroprotective effect of SA against IR injury in rats, the control group was not stable in the experiment of studying the activity of SA. Thus, I could not judge whether SA was indeed inactive. I suggest the author to add experiments to detect whether the SA has an effect on the concentration of HIF-1alpha, to reflect the characteristics of SA from this side.

Minor comments:
1. “Figure 3 and 5”. Missing the scale in the pathological images.
2. “Abstract”. Missing the “Conclusion”.

Reviewer 3 ·

Basic reporting

The manuscript by Eduardo Cienfuegos-Pecina et al. ‘Sodium (S)-2-hydroxyglutarate, but not succinic acid, has a nephroprotective effect against ischemia-reperfusion injury in Wistar rats’, suggests a role of the HIF-1 pathway’, dealing with the ameliorative effect of (S)-2-hydroxyglutarate in renal IR injury in Wistar rats.
The study objectives are clear, and experiments are nicely done.
Author should add some latest research/references. Which are lacking in text. Like discuss some basic signaling related to HIF pathways which may be effected by S)-2-hydroxyglutarate and succinic acid.

Experimental design

Authors performed many experiments and assessed all the basic parameters with good stats.

Validity of the findings

no comments

Additional comments

Authors need to justify following comments and some of which should be included in the discussion of the manuscript so that readers can easily understand.
Comments:
1. Author should justify why male rats were used for assessing (S)-2-hydroxyglutarate while female for succinic acid?
2. Authors should justify why they use rat IR model? Whether rats are good for IR models than mice? What are the metabolic changes induced by HIF 1 accumulation in these models? (S)-2-hydroxyglutarate metabolism in rat vs mice system? Which model is more relevant to humans?

3. How would authors justify the doses of (S)-2-hydroxyglutarate which ameliorates the effects of renal IR injury but what about its other potential molecular effects like activation of oncogene or inactivation of tumor suppressor genes in the long term? Such effects are already established to be induced by high concentration of (S)-2-hydroxyglutarate.

4. As authors previously published same type of study on plant extract ‘Nephroprotective Effect of Sonchus oleraceus Extract against Kidney Injury Induced by Ischemia-Reperfusion in Wistar Rats’, I hope they are wellaware of using different compounds in the protection renal IR injury as well as the longterm effects of those compounds. Please justify my point.

5. Moreover, Authors only assessed kidney tissue but what about the effects of (S)-2-hydroxyglutarate and succinic acid in other organs? We should be cautious when giving non targeted compounds by orally.

6. What is the conversion rate of (S)-2-hydroxyglutarate to alpha ketoglutarate?

7. Is it possible that (S)-2-hydroxyglutarate treatment may change the epigenetic pattern of DNA viz; hypermethylation of DNA and cause tumor in long term effect? Because in this study all rats were sacrificed within 24 hrs only following start of treatment schedule.

8. Is it possible that tramadol induces some immunomodulatory effects or neutralizes the immunomodulatory effects of IR / (S)-2-hydroxyglutarate?

9. In Abstract methods section line 37 … (S)-2HG and SA were independently evaluated in rats after renal IR. It should be ‘renal IR injury’.

10. In the results, the authors report that 12.5 tox and IR groups show HIF 1 alpha accumulation but not in 12.5+IR and 25+IR groups. Can authors clarify in the manuscript whether these are negative results as per the hypothesis in terms of HIF1 alpha accumulation.

11. At some places authors write inhibition of EGLN and at others as inhibition of prolyl-4-hydroxylases. They should either write EGLN family member prolyl-4-hydroxylases or only inhibition of prolyl-4-hydroxylases at all the places. I suggest it would be best if they focus on what has been analyzed in their experiments.

12. Please write H1F1 alpha or H1F1 α at all places throughout text to maintain uniformity.

Line 32 in abstract please write renal IR injury.

Line 48 result section please write SA did not ‘show’ instead of ‘have a’ nephroprotective effect

Line 415 Discussion…(S)-2HG caused no toxic effects at the hepatic or renal level, as shown by the normal values for the biochemical markers…..should be… as shown by the normal values ‘of’ biochemical markers.

Line 455-456 Discussion…. (S)-2HG caused a significant accumulation of HIF-1α in the kidney tissue in the nontoxicity groups of rats. Please write….(S)-2HG caused a significant accumulation of HIF-1α in the kidney tissue of rats in nontoxicity groups.

Line 464 concentration in the 12.5+IR and 25+IR groups in the late phase in our study Please write ………. in the late/reperfusion phase of this study.

---

## Round 0.2 · accepted · Accept

Thank you for revising your original manuscript.

·

Basic reporting

No comment.

Experimental design

No comment.

Validity of the findings

No comment.

Additional comments

Kidney transplantation is the definitive treatment for patients with end-stage chronic kidney disease. However, the shortage of available organs for transplantation is a formidable challenge. Ischemia–reperfusion (IR) injury is the main cause of delayed graft function in solid organ transplantation. It is inspiriting and beneficial to analyze the activities and potential pharmacological properties of preconditioning drugs (hypoxia mimetics) on preventing or limitting IR injury. In this manuscript, Dr. Eduardo Cienfuegos-Pecina and colleagues looked at the nephroprotective effect of the egg-laying defective nine homolog (EGLN) inhibitors, (S)-2-hydroxyglutarate [(S)-2HG] and succinic acid (SA) in IR injury rat. The study’s conclusion showed that only (S)-2HG showed a dose-dependent nephroprotective effect at the evaluated doses, which involved the promotion of HIF-1alpha accumulation. The paper is improved and most concerned raised by the reviewer have been addressed. I think it is might suitable for publication at this version of revised manuscript.